# MLLMEraser: Achieving Test-Time Unlearning in Multimodal Large Language Models through Activation Steering

## Abstract

Multimodal large language models (MLLMs) have demonstrated remarkable capabilities across vision–language tasks, yet their large-scale deployment raises pressing concerns about memorized private data, outdated knowledge, and harmful content. Existing unlearning approaches for MLLMs typically adapt training-based strategies such as gradient ascent or preference optimization, but these methods are computationally expensive, irreversible, and often distort retained knowledge. In this work, we propose MLLMEraser, an input-aware, training-free framework for test-time unlearning. Our approach leverages activation steering to enable dynamic knowledge erasure without parameter updates. Specifically, we construct a multimodal erasure direction by contrasting adversarially perturbed, knowledge-recall image–text pairs with knowledge-erasure counterparts, capturing both textual and visual discrepancies. To prevent unnecessary interference, we further design an input-aware steering mechanism that adaptively determines when and how the erasure direction should be applied, preserving utility on retained knowledge while enforcing forgetting on designated content. Experiments on LLaVA-1.5 and Qwen-2.5-VL demonstrate that MLLMEraser consistently outperforms state-of-the-art MLLM unlearning baselines, achieving stronger forgetting performance with lower computational cost and minimal utility degradation.

## 1 Introduction

Multimodal large language models (MLLMs) (Liu et al., 2023; Wang et al., 2024; Zhu et al., 2024; Yang et al., 2023; Anil et al., 2023) have demonstrated remarkable capabilities in tasks such as visual question answering (Hu et al., 2024; Kuang et al., 2025), image–text generation (Wu et al., 2024b; Lan et al., 2025), and embodied AI applications (Wu et al., 2024a; Cheng et al., 2025). However, their large-scale deployment raises concerns about memorizing problematic information once learned, particularly in privacy-sensitive (Huo et al., 2025) or safety-critical (Liu et al., 2024c) applications, highlighting the need for reliable unlearning mechanisms to ensure trustworthy MLLM systems (Li et al., 2024; Liu et al., 2025c; Chen et al., 2025). MLLM unlearning aims to selectively erase designated information across modalities while preserving general utility, thereby supporting privacy protection, mitigating misuse, and maintaining reliability (Dontsov et al., 2025). Existing work mainly adapts training–based strategies from LLM unlearning, employing gradient ascent, preference optimization (Zhang et al., 2024a), or performing targeted parameter updates (Liu et al., 2025d; Huo et al., 2025). While effective, these training-based methods introduce substantial computational costs, inference latency, and risks of corrupting retained knowledge (Ding et al., 2025). This motivates test-time MLLM unlearning (See Figure 1), a paradigm that prevents the generation of designated information at inference without modifying model parameters, which offers an immediate, lightweight, and reversible solution.

Recently, activation steering (Turner et al., 2023; Wang et al., 2025a) emerges as a promising approach for test-time intervention. Activation steering manipulates the internal computation of LLMs by injecting a carefully constructed direction vector into their intermediate activations, shifting the model's latent representation toward a desired semantic space and inducing specific behaviors or responses. However, existing studies on steering have focused mainly on safety alignment (Sheng et al., 2025a; Zhao et al., 2025), reasoning length regulation (Sun et al., 2025; Sheng et al., 2025b),

Figure 1: (a) Comparison between training-based and test-time unlearning paradigms for MLLMs. (b) Illustration of the activation steering process. (c)–(d) Differences between existing methods and ours in constructing and applying the steering vector.

and hallucination reduction (Liu et al., 2025a; Wang et al., 2025b), leaving its potential for test-time MLLM unlearning largely unexplored. Here we aim to leverage the superiority of activation steering for test-time MLLM unlearning, yet encounter two fundamental challenges: multimodal erasure direction construction and multimodal erasure direction application.

- **Multimodal erasure direction construction:** Multimodal erasure direction construction denotes how to construct an effective activation steering vector for MLLMs. Traditional steering methods can directly elicit contrastive activation pairs from existing models by prompting for different response types, such as truthful versus deceptive answers (Wang et al., 2025c) or safe versus unsafe outputs (Arditi et al., 2024). However, in the context of MLLM unlearning, the ideal contrastive samples would come from models that have already forgotten the target information versus those that retain it—but such "unlearned" models are precisely what we seek to avoid obtaining through expensive retraining. Moreover, given that MLLMs inherently encode joint visual-textual representations through deep cross-modal fusion (Liu et al., 2023; Bai et al., 2025), existing steering works that predominantly rely on textual contrasts (Gan et al., 2025) while neglecting visual signal processing produce incomplete erasure directions, leading to insufficient forgetting performance.

- **Multimodal erasure direction application:** Multimodal erasure direction application focuses on when to selectively apply this direction. Even with a well-extracted multimodal erasure direction, deciding when to apply it remains an open challenge. Current steering methods typically apply uniform interventions across all inputs (Turner et al., 2023; Arditi et al., 2024), which can effectively mitigate unsafe or privacy-leaking outputs but frequently distorts responses to non-targeted queries and degrades performance on retained knowledge (Lee et al., 2025; Zhao et al., 2025). An effective unlearning approach requires input-aware activation that selectively triggers interventions only for content requiring unlearning (*i.e.,* forget set) while preserving normal model behavior for retained information (*i.e.,* retain set). However, unlike scenarios with clear semantic distinctions (*e.g.,* positive vs. negative sentiment), forget and retain examples often share identical formats and content types—both may involve user-attribute queries that differ only in their unlearning designation (Liu et al., 2025c). Such similarity makes accurate selective control difficult and increases the risk of over-forgetting (Xu et al., 2025).

To address above challenges, we introduce MLLMEraser, an input-aware test-time MLLM unlearning framework that leverages activation steering for dynamic and test-time information erasure without parameter modification. For the multimodal erasure direction construction, we ground the response-intervention objective of unlearning in the model's intrinsic refusal behavior and derive the erasure direction by contrasting the semantics of knowledge-recall and knowledge-erasure. Specifically, we generate two contrastive sets to extract the direction: a negative set (*i.e.,* knowledge-recall inputs) combining jailbreak prompts with adversarially perturbed images to induce unsafe or privacy-sensitive outputs, and a positive set (*i.e.,* knowledge-erasure inputs) that elicits refusal-style response (*e.g.,* "I cannot answer this question.") with the corresponding clean images. After obtaining the steering vector, we reformulate steering as an input-aware task by introducing a direction-

determining function $f(\cdot)$, rather than applying the direction indiscriminately. Given the hidden activations $\mathbf{h}$, this function adaptively decides the steering vector $f(\mathbf{h})$. For forget data, $f(\mathbf{h})$ maps the activations towards the pre-computed erasure direction, while for retain data, $f(\mathbf{h})$ degenerates into a null direction (Fang et al., 2025a), yielding nearly zero intervention and leaving the representation distribution unchanged. Inspired by (Sheng et al., 2025a), we implement $f(\mathbf{h})$ as a simple yet effective linear transformation, circumventing the need for additional auxiliary model training. Experiments on LLaVA-1.5 (Liu et al., 2023) and Qwen-2.5-VL (Bai et al., 2025) demonstrate the effectiveness of MLLMEraser, which consistently outperforms state-of-the-art MLLM unlearning methods while providing an efficient and lightweight solution that balances unlearning performance with model utility preservation.

## 2 PRELIMINARY

This section begins by introducing the notation and formalizing the problem of test-time MLLM unlearning in Section 2.1. Section 2.2 outlines how activation steering offers a feasible mechanism for test-time MLLM unlearning , enabling test-time behavior control without parameter updates.

### 2.1 NOTATION AND PROBLEM SETUP

Given a multimodal instruction-tuning dataset $\mathcal{D} = \{(\mathcal{I}_i, \mathcal{Q}_i, \mathcal{A}_i)\}_{i=1}^{N}$ of size $N$, where $\mathcal{I}_i$ denotes the input image, $\mathcal{Q}_i$ is the textual instruction, and $\mathcal{A}_i = (y_1^{(i)}, y_2^{(i)}, \ldots, y_{|\mathcal{A}_i|}^{(i)})$ represnets the target answer sequence, the MLLM is fine-tuned to maximize the likelihood of predicting each token $y_t$ given the multimodal context and the previously generated tokens. Specifically, the image is first encoded by a vision encoder and projected into the language space through a multimodal adapter. This fused representation is then concatenated with the tokenized query and processed autoregressively by the LLM backbone to generate the answer (Liu et al., 2023). The optimization objective for the MLLM model parameterized by $\theta$ can be written as follows:

$$\min_{\theta} - \sum_{i=1}^{N} \sum_{t=1}^{|\mathcal{A}_i|} \log P_{\theta}(y_t^{(i)} \mid \mathcal{I}_i, \mathcal{T}_i, y_{<t}^{(i)}), \qquad (1)$$

where $y_{<t}^{(i)} = (y_1^{(i)}, \ldots, y_{t-1}^{(i)})$ represents tokens preceding $y_t^{(i)}$. In the MLLM unlearning setting, the dataset is partitioned into two disjoint subsets: the forget set $\mathcal{D}_f = \{(\mathcal{I}_i, \mathcal{Q}_i, \mathcal{A}_i)\}_{i=1}^{N_f}$, where the model should not recall or answer queries, and the retain set $\mathcal{D}_r = \mathcal{D} \setminus \mathcal{D}_f = \{(\mathcal{I}_j, \mathcal{Q}_j, \mathcal{A}_j)\}_{j=1}^{N_r}$, on which the model is expected to preserve its utility after unlearning.

**Training–based MLLM unlearning.** This paradigm aims to obtain an unlearned model parameterized by $\hat{\theta}$ via jointly optimizing the forget loss $\mathcal{L}_f$ and the retain loss $\mathcal{L}_r$, formulated as:

$$\arg\min_{\hat{\theta}} \ \lambda_f \, \mathbb{E}_{(\mathcal{I}_i, \mathcal{Q}_i, \mathcal{A}_i) \sim \mathcal{D}_f} \big[ \mathcal{L}_f(\mathcal{I}_i, \mathcal{Q}_i, \mathcal{A}_i; \hat{\theta}) \big] \ + \ \lambda_r \, \mathbb{E}_{(\mathcal{I}_j, \mathcal{Q}_j, \mathcal{A}_j) \sim \mathcal{D}_r} \big[ \mathcal{L}_r(\mathcal{I}_j, \mathcal{Q}_j, \mathcal{A}_j; \hat{\theta}) \big], \quad (2)$$

where $\lambda_f, \lambda_r$ are trade-off parameters. The retain loss $\mathcal{L}_r$ is commonly instantiated as an autoregressive negative log-likelihood (NLL) or a KL-divergence constraint term (Liu et al., 2025c), ensuring that the model preserves performance on $\mathcal{D}_r$. The forget loss $\mathcal{L}_f$ serves as the unlearning objective, typically implemented through gradient ascent (Thudi et al., 2022) or preference-based optimization (Zhang et al., 2024a), encouraging the model to deviate from its original predictions.

**Test-time MLLM unlearning.** In this setting, the goal is to discourage the model from producing $\mathcal{A}_i$ during inference while keeping the parameter $\theta$ fixed. The objective is to prevent the model from recalling or generating undesired knowledge associated with the forget set $\mathcal{D}_f$—for instance, by producing incorrect answers or refusal-style responses—by means of test-time intervention rather than parameter updates. At the same time, responses to non-target inputs from the retain set $\mathcal{D}_r$ are expected to remain unaffected, ensuring that the model preserves its normal capabilities while selectively unlearning only the designated content.

Figure 2: Overview of the proposed MLLMEraser framework. Stage 1 derives a multimodal erasure direction $\mathbf{d}_{\text{erase}}$ from contrastive image-text pairs. Stage 2 introduces an input-aware steering mechanism $f(\mathbf{h})$ that adaptively applies $\mathbf{d}_{\text{erase}}$ to shift the activations of forget samples toward refusal-style responses, while leaving retain samples nearly unaffected to preserve correct responses.

## 2.2 Behavioral Control through Activation Steering

Activation steering has recently been explored as an effective way to modulate model behavior at inference time without modifying parameters. In the context of unlearning, the steering vector is referred to as the erasure direction, which captures the representational shift between knowledge-recall samples and their knowledge-erasure counterparts. Formally, let $\mathbf{h}^\ell \in \mathbb{R}^d$ denote the hidden activation at layer $\ell$, the erasure direction $\mathbf{d}_{\text{erase}} \in \mathbb{R}^d$ is commonly estimated using the difference-in-means between the hidden activations of the two contrastive groups:

$$\mathbf{d}_{\text{erase}} = \frac{1}{|\mathcal{D}^+|} \sum_{(\mathcal{I},\mathcal{Q}) \in \mathcal{D}^+} \mathbf{h}^\ell(\mathcal{I},\mathcal{Q}) - \frac{1}{|\mathcal{D}^-|} \sum_{(\mathcal{I},\mathcal{Q}) \in \mathcal{D}^-} \mathbf{h}^\ell(\mathcal{I},\mathcal{Q}), \tag{3}$$

where $\mathcal{D}^+$ denotes the set of knowledge-erasure samples and $\mathcal{D}^-$ the corresponding knowledge-recall samples, which will be discussed in Section 3.1. The resulting direction is subsequently added to the hidden states to steer the representation, formulated as:

$$\tilde{\mathbf{h}}^\ell = \mathbf{h}^\ell + \lambda \cdot \mathbf{d}_{\text{erase}}, \tag{4}$$

where $\lambda \in \mathbb{R}$ controls the strength of the adjustment. For simplicity, we omit the layer superscript $\ell$ in subsequent notation. By applying this operation to selected layers, the model's outputs on the forget set are steered away from generating privacy-sensitive and unsafe responses.

## 3 MLLMEraser

We propose MLLMEraser, an input-aware test-time unlearning framework for MLLMs based on activation steering. In Section 3.1, we detail the construction of the multimodal erasure direction from knowledge-recall and knowledge-erasure text–image pairs. We present the input-aware mechanism that selectively applies the erasure direction at inference time in Section 3.2, and conclude the complete framework in Section 3.2.

### 3.1 Multimodal Erasure Direction Construction

Inspired by recent research on LLM and MLLM safety (Shao et al., 2024; Liu et al., 2024a; Fang et al., 2025b), in which aligned models can refuse to answer harmful queries (*e.g.,* "I cannot provide this information"), we observe that such intrinsic refusal behavior is conceptually consistent with the goal of test-time unlearning—the erasure of target information at the response level. In fact, answering a query with relevant knowledge naturally corresponds to the process of knowledge-recall, whereas refusing to respond aligns with knowledge-erasure. Building on this insight, we leverage the model's inherent refusal capacity to facilitate more flexible response intervention and derive the erasure direction by contrasting the semantics of knowledge-recall and knowledge-erasure. Specifically, we construct two types of harmful prompts to capture the refusal behavior: (1) rejected harmful inputs $\mathcal{Q}_i$ (*i.e.,* knowledge-erasure prompts), which trigger refusal behavior; and (2) complied harmful inputs $\mathcal{Q}_i'$ (*i.e.,* harmful knowledge-recall prompts), which bypass safety mechanisms and elicit malicious outputs. The textual erasure direction then can be derived by computing the activation difference between these two contrastive groups (Arditi et al., 2024), as detailed in Equation 3. By

leveraging the model's refusal behavior, our approach constructs the erasure direction independently of activations from pre- and post-unlearning samples, obviating the need for an unlearned model.

However, constructing erasure direction from textual contrastive pairs alone is ineffective for MLLM unlearning. As visual embeddings are projected into the LLM's semantic space and fused with text via attention, resulting joint cross-modal representations (Liu et al., 2023; Wang et al., 2024). Recent research has shown that the visual modality introduces a new attack surface, where adversarial images paired with harmful instructions can induce MLLMs to generate malicious content (Qi et al., 2024). These adversarial inputs exploit the model's knowledge-recall capability to elicit sensitive responses, whereas a clean image paired with a rejected instruction reflects knowledge erasure, in which the model suppresses the targeted information instead of recalling it. Inspired by this, we generate perturbed images that maximize the probability of harmful responses, incorporating visual information and amplifying the model's tendency toward harmful knowledge-recall behavior. Specifically, given a rejected harmful instruction $\mathcal{Q}_i$ and a clean image $\mathcal{I}_i$, we construct the adversarially perturbed image $\mathcal{I}_i'$ to elicit harmful knowledge-recall behavior by solving the following optimization problem (Qi et al., 2024; Madry et al., 2017):

$$\mathcal{I}_i' := \arg\max_{\mathcal{I} \in \mathcal{B}} \sum_{y \in \mathcal{Y}_f} \log P_\theta \left(y \mid \mathcal{I}, \mathcal{Q}_i\right), \tag{5}$$

where $\mathcal{Y}_f$ is a small few-shot corpus of harmful target responses, $\mathcal{B} = \{\mathcal{I} \mid \|\mathcal{I} - \mathcal{I}_i\|_p \leq \varepsilon\}$ is some constraint applibuxiaed to the input space bounded by $\ell_p$ norm, and $\varepsilon$ controls the perturbation budget. We use a multi-step projected gradient descent (PGD) update algorithm (Madry et al., 2017) to generate the adversarial image $\mathcal{I}_i'$. The update rule at step $k+1$ is as follows:

$$\mathcal{I}_i^{(k+1)'} = \Pi_{\mathcal{I}_i + \mathcal{B}}\left(\mathcal{I}_i^{(k)'} + \alpha \cdot \text{sign}\left(\nabla_\mathcal{I} \log P_\theta\left(y \mid \mathcal{I}_i^{(k)'}, \mathcal{Q}_i\right)\right)\right), \tag{6}$$

where $\alpha$ is the step size, and $\Pi$ denotes the projection operator that maps the updated sample back onto the feasible set $\mathcal{B}$. Here, $\text{sign}(\cdot)$ denotes the element-wise sign function, *i.e.,* $\text{sign}(x) = +1$ if $x \geq 0$ and $-1$ otherwise. Finally, we can obtain two contrastive sets of image-text pairs: (1) harmful knowledge-recall pairs, composed of adversarial images $\mathcal{I}'$ paired with harmful instructions $\mathcal{Q}'$ that induce malicious knowledge, forming the negative set $\mathcal{D}^- = \{(\mathcal{I}_i', \mathcal{Q}_i')\}_{i=1}^N$; and (2) knowledge-erasure pairs, consisting of clean images $\mathcal{I}$ paired with rejected harmful prompts $\mathcal{Q}$ that elicit the model's refusal behavior and achieve response-level knowledge erasure, thereby forming the positive set $\mathcal{D}^+ = \{(\mathcal{I}_i, \mathcal{Q}_i)\}_{i=1}^N$. Then the multimodal erasure direction can be calculated as:

$$\mathbf{d}_{\text{erase}}^\ell = \frac{1}{|\mathcal{D}^+|} \sum_{(\mathcal{I}, \mathcal{Q}) \in \mathcal{D}^+} \mathbf{h}(\mathcal{I}, \mathcal{Q}) - \frac{1}{|\mathcal{D}^-|} \sum_{(\mathcal{I}', \mathcal{Q}') \in \mathcal{D}^-} \mathbf{h}(\mathcal{I}', \mathcal{Q}'). \tag{7}$$

By exploiting the model's intrinsic refusal behavior, our design derives a multimodal erasure direction that enforces response-level knowledge erasure on the forget set, enabling refusal-oriented interventions and achieving effective unlearning.

## 3.2 INPUT-AWARE STEERING WITH ERASURE DIRECTIONS

After getting the multimodal erasure direction, it is essential to determine when to apply it, so that it would not affect the model performance on the retain set. Current steering methods often indiscriminately apply the steering vector to all prompts (Rimsky et al., 2024), a strategy that inevitably degrades overall model performance. Regarding unlearning, this degradation manifests as corrupted responses to samples in the retain set, leading to over-forgetting problem (Xu et al., 2025). An desirable unlearning steering mechanism should be input-aware: for samples in the forget set, their activations should be steered toward the knowledge-erasure behavior, while for samples in the retain set, their activations should remain as unchanged as possible. To achieve this, the steering task can be formulated as an input-aware task by constructing a direction-determining function $f(\mathbf{h}(\mathcal{I}, \mathcal{Q}))$, which conditions on the query's activation and produces the steering direction applied at inference. Then the steering process can be written as:

$$\tilde{\mathbf{h}} = \mathbf{h} + \lambda f(\mathbf{h}). \tag{8}$$

More specific, the function can be formally expressed as follows:

$$f(\mathbf{h}(\mathcal{I}, \mathcal{Q})) \approx \begin{cases} \mathbf{d}_{\text{erase}}, & \text{if } (\mathcal{I}, \mathcal{Q}) \in \mathcal{D}_f, \\ \mathbf{0}, & \text{if } (\mathcal{I}, \mathcal{Q}) \in \mathcal{D}_r. \end{cases} \tag{9}$$

Inspired by (Sheng et al., 2025a), we implement $f(\mathbf{h})$ as a linear transformation, given by $f(\mathbf{h}) = \mathbf{W}\mathbf{h}$, where $\mathbf{W} \in \mathbb{R}^{d \times d}$. The optimization objective can then be naturally formulated as the following constrained least-squares problem:

$$\arg \min_{\mathbf{W}} \left( \|\mathbf{W}\mathbf{H}_f - \mathbf{D}\|^2 + \gamma \|\mathbf{W}\|^2 \right), \quad \text{s.t. } \mathbf{W}\mathbf{H}_r = \mathbf{0}, \tag{10}$$

where $\mathbf{H}_f \in \mathbb{D}^{d \times N_f}$ and $\mathbf{H}_r \in \mathbb{R}^{d \times N_r}$ denote the activation matrices obtained from the last token of prompts in the forget set $\mathcal{D}_f$ and the retain set $\mathcal{D}_r$, respectively. Here $\|\cdot\|$ denotes the Frobenius norm, while $\gamma$ is a regularization hyper-parameter, and $\mathbf{D} \in \mathbb{R}^{d \times N_f}$ is formed by stacking $N_f$ identical copies of the same multimodal erasure direction vector column-wise.

For preserving the model performance on the retain set, we constrain the direction-determining function with null-space constraints (Sheng et al., 2025a; Fang et al., 2025a). In particular, if a matrix $\mathbf{B}$ lies in the left null space of $\mathbf{A}$, it satisfies $\mathbf{B}\mathbf{A} = \mathbf{0}$ (Dieudonne, 1969). Motivated by this property, we project $\mathbf{W}$ into the null space of $\mathbf{H}_r$ through a projection matrix $\mathbf{P}$ and optimize the projected matrix $\mathbf{W}\mathbf{P}$. Since the left null space of $\mathbf{H}_r$ is equivalent to that of positive semidefinite matrix $\mathbf{H}_r\mathbf{H}_r^\top \in \mathbb{R}^{d \times d}$ (see the proof in Appendix B), we first apply a Singular Value Decomposition (SVD) to $\mathbf{H}_r\mathbf{H}_r^\top$ to calculate the projection matrix as: $\{\mathbf{U}, \mathbf{\Sigma}, \mathbf{U}^\top\} = \mathrm{SVD}(\mathbf{H}_r\mathbf{H}_r^\top)$, where $\mathbf{U} \in \mathbb{R}^{d \times d}$ is an orthogonal matrix whose columns are the eigenvectors of $\mathbf{H}_r\mathbf{H}_r^\top$, and $\mathbf{\Sigma} \in \mathbb{R}^{d \times d}$ is a diagonal matrix containing its singular values. We can then partition the eigenvector matrix $\mathbf{U}$ into two sub-matrices: $\mathbf{U}_1 \in \mathbb{R}^{d \times k}$ and $\mathbf{U}_2 \in \mathbb{R}^{d \times (d-k)}$. The columns of $\mathbf{U}_1$ correspond to the non-zero singular values, spanning the column space of $\mathbf{H}_r\mathbf{H}_r^\top$. Conversely, the columns of $\mathbf{U}_2$ are the eigenvectors corresponding to the zero eigenvalues, which form a orthonormal basis for the null space of $\mathbf{H}_r\mathbf{H}_r^\top$. The projection matrix $\mathbf{P}$ can be given by $\mathbf{P} = \mathbf{U}_2\mathbf{U}_2^\top$. The projected matrix $\mathbf{W}\mathbf{P}$ naturally lies in the null space of $\mathbf{H}_r\mathbf{H}_r^\top$ and satisfies $\mathbf{W}\mathbf{P}\mathbf{H}_r\mathbf{H}_r^\top = \mathbf{W}\mathbf{P}\mathbf{H}_r = \mathbf{0}$. Then, the optimization objective in Equation 10 can be rewritten as:

$$\mathbf{W}^* := \arg \min_{\mathbf{W}} \left( \|\mathbf{W}\mathbf{P}\mathbf{H}_f - \mathbf{D}\| + \gamma \|\mathbf{W}\mathbf{P}\| \right). \tag{11}$$

The closed-form solution of Equation 11 can be given by:

$$\mathbf{W}^* = \mathbf{D}\mathbf{H}_f^\top \mathbf{P}^\top \left( \mathbf{P}\mathbf{H}_f\mathbf{H}_f^\top \mathbf{P}^\top + \gamma \mathbf{P}\mathbf{P}^\top \right)^+, \tag{12}$$

where $^+$ represents the pseudoinverse. In this way, we construct an input-aware mapping mechanism $f(\mathbf{h}) = \mathbf{W}\mathbf{P}\mathbf{h}$. This mechanism ensures that for forget data, $f(\mathbf{h})$ maps activations toward the extracted multimodal erasure direction, whereas for retain data, it collapses to nearly zero vector, leaving the representation distribution unchanged.

## 3.3 Final Formulation

After integrating (1) construction of multimodal erasure directions $\mathbf{d}_{\text{erase}}$ and (2) an input-aware steering mechanism $f(\mathbf{h})$ in a unified pipeline. The final steering process of MLLMEraser is formulated as:

$$\tilde{\mathbf{h}} = \mathbf{h} + \lambda f(\mathbf{h}) = \mathbf{h} + \lambda \mathbf{W}\mathbf{P}\mathbf{h}, \tag{13}$$

By selectively steering activations toward the erasure direction at inference, MLLMEraser achieves test-time unlearning with a favorable trade-off between unlearning performance and model utility.

## 4 Experiment

This section provides an extensive experimental evaluation of MLLMEraser, with the analysis structured around answering the following key research questions: **RQ1:** How does MLLMEraser perform *w.r.t.* forget quality and model utility? **RQ2:** What is the impact of multimodal erasure direction and input-aware erasure direction application on unlearning performance? **RQ3:** How does the efficiency of MLLMEraser compare to other unlearning methods?

### 4.1 Experimental Setups

We use LLaVA-1.5-7B (Liu et al., 2023) and Qwen-2.5-VL-7B-Instruct (Bai et al., 2025) as the MLLM backbones and evaluate on the widely adopted unlearning benchmark MLLMU-Bench (Liu

Table 1: Unlearning performance on MLLMU-Bench (5% Forget). Results are evaluated on the forget set (Fgt), test set (Test), retain set (Ret), and celebrity set (Cele). ↓ indicates lower is better, and ↑ indicates higher is better. The best results are highlighted in bold.

| Models | Classification Accuracy (%) | | | | Generation: Rouge Score | | | | Cloze: Accuracy (%) | | | |
|---|---|---|---|---|---|---|---|---|---|---|---|---|
| | Fgt↓ | Test↓ | Ret↑ | Cele↑ | Fgt↓ | Test↓ | Ret↑ | Cele↑ | Fgt↓ | Test↓ | Ret↑ | Cele↑ |
| LLaVA-1.5-7B (5% Forget)-VQA | | | | | | | | | | | | |
| Vanilla | 42.40 | 40.00 | 45.23 | 47.00 | 0.632 | 0.286 | 0.585 | 0.312 | 54.00 | 24.00 | 55.37 | 8.17 |
| GA | 40.08 | 38.40 | 39.75 | 38.77 | 0.353 | 0.251 | 0.389 | 0.265 | 14.00 | 16.00 | 28.00 | 5.56 |
| GA_Diff | 35.20 | 29.60 | 36.00 | 37.10 | 0.554 | 0.265 | 0.585 | 0.310 | 32.00 | 20.00 | 52.11 | 7.84 |
| KL_Min | 40.00 | 36.80 | 42.87 | 44.78 | 0.629 | 0.309 | 0.585 | 0.307 | 52.00 | 18.00 | 53.58 | 6.86 |
| NPO | 32.80 | 39.20 | 36.16 | 43.60 | 0.360 | 0.243 | 0.363 | 0.278 | 20.00 | 16.00 | 23.89 | 4.90 |
| MMUnlearner | 33.60 | 33.60 | 40.93 | 41.12 | 0.530 | 0.271 | 0.551 | 0.310 | 28.00 | 20.00 | 44.84 | 5.56 |
| Ours | **11.20** | **25.20** | **43.54** | **45.43** | **0.106** | **0.209** | **0.592** | **0.311** | **6.00** | **14.00** | **54.21** | **8.17** |
| Qwen-2.5-VL-7B (5% Forget)-VQA | | | | | | | | | | | | |
| Vanilla | 67.20 | 68.80 | 66.20 | 78.33 | 0.642 | 0.317 | 0.583 | 0.453 | 36.00 | 22.00 | 40.42 | 26.80 |
| GA | 64.00 | 64.80 | 65.11 | 74.15 | 0.419 | 0.315 | 0.438 | 0.335 | 16.00 | 14.00 | 16.21 | 25.16 |
| GA_Diff | 56.00 | 59.20 | 64.05 | 76.76 | 0.482 | 0.296 | 0.566 | 0.442 | 16.00 | 22.00 | 33.79 | 24.84 |
| KL_Min | 51.20 | 50.40 | 50.30 | 48.57 | 0.594 | 0.322 | 0.578 | 0.441 | 32.00 | 18.00 | 34.32 | 25.16 |
| NPO | 54.40 | 59.20 | 56.08 | 74.08 | 0.492 | 0.273 | 0.492 | 0.429 | 12.00 | 18.00 | 23.26 | 23.52 |
| MMUnlearner | 60.00 | 58.40 | 53.84 | 72.45 | 0.491 | 0.270 | 0.538 | 0.433 | 12.00 | 18.00 | 31.79 | 22.86 |
| Ours | **19.20** | **38.40** | **65.86** | **78.32** | **0.165** | **0.175** | **0.580** | **0.445** | **10.00** | **12.00** | **39.37** | **26.47** |

et al., 2025c), which centers on fictitious profiles at both visual and textual levels. This benchmark includes four datasets: the Forget Set (fictitious profiles designated for unlearning), the Test Set (paraphrased and image-transformed variants for generalization), the Retain Set (fictitious profiles that should be preserved), and the Real Celebrity Set (real-world profiles for utility evaluation). It further defines three tasks: classification, generation, and cloze, which are evaluated with classification accuracy, ROUGE-L score (Lin, 2004), and cloze accuracy, respectively. Comprehensive details on the benchmark, baselines, evaluation metrics, and implementation of MLLMEraser are provided in Appendix D.

## 4.2 Results Analysis on MLLM Unlearning (RQ1)

We evaluate several MLLM unlearning methods on four datasets: the forget and test sets assess unlearning performance, while the retain and celebrity sets evaluate model utility. Table 1 presents results on classification, generation, and cloze tasks, and Figure 3 illustrates the trade-off between forget quality and model utility across different forget ratios, where points closer to the upper-right corner indicate better balance. From these results, we can draw the following observations:

- **MLLMEraser demonstrates consistently superior unlearning efficacy across all tasks.** Specifically, it degrades performance on the forget set by an average of 39.6% in classification accuracy, 37.0% in cloze accuracy, and 0.502 in ROUGE-L score compared with vanilla models across two MLLM backbones, underscoring the effectiveness of our test-time unlearning approach. In contrast, training-based methods rely on the limited supervision signals from the forget set to update model parameters, which often results in incomplete forgetting.

- **MLLMEraser effectively preserves the retained knowledge.** In particular, it remains closest to the vanilla models on the retain and celebrity sets, with only 1.63% deviations in classification accuracy, 0.17% in cloze accuracy, and 0.002 in ROUGE-L under both backbones, which can be attributed to the strength of our input-aware steering mechanism in safeguarding retained knowledge. Instead, training-based methods rely on the retain set to constrain the unlearned model's output distribution to match that of the original model. While partially effective, this parameter-update paradigm inevitably degrades overall performance.

- **MLLMEraser achieves the best trade-off between unlearning performance and model utility.** As shown in Figure 3, our method (top-right corner) consistently achieves substantial performance reductions on the forget set, with only minor drops on retained knowledge. In fact, training-based methods suffer from gradient conflicts between the forget and retain sets, which complicates parameter updates and prevents them from maintaining a favorable balance. By selectively intervening at test time—without any parameter updates—our approach circumvents this conflict, enabling effective unlearning while preserving overall performance.

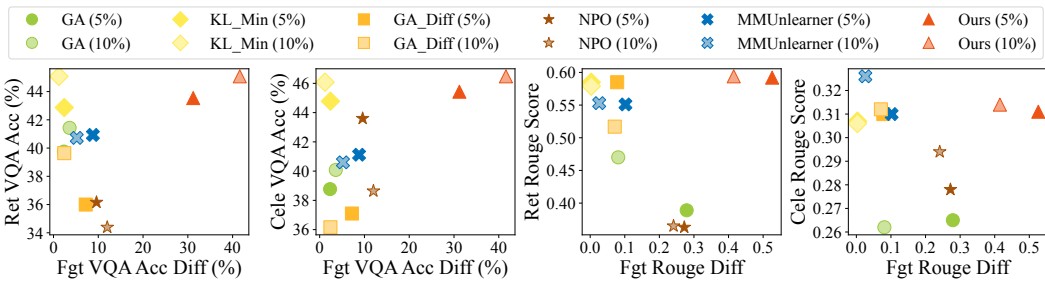

Figure 3: Trade-off between forget quality and model utility on LLaVA under 5% and 10% forget ratios. The left two plots correspond to classification task, where the $x$-axis shows accuracy difference on the forget set (Fgt VQA Acc Diff), and the right two plots correspond to generation, where the $x$-axis shows ROUGE-L difference on the forget set (Fgt Rouge Diff). The $y$-axis reports model utility on the retained (Ret) and celebrity (Cele) sets.

Table 2: Ablation study on MLLMU-Bench (5% Forget) using the Qwen-2.5-VL-7B-Instruct model. Results are reported on the forget set (Fgt), test set (Test), retain set (Ret), and celebrity set (Cele). ↓ indicates lower is better, and ↑ indicates higher is better.

| Models | Classification: Accuracy (%) | | | | Generation: Rouge Score | | | | Cloze: Accuracy (%) | | | |
|---|---|---|---|---|---|---|---|---|---|---|---|---|
| | Fgt ↓ | Test ↓ | Ret ↑ | Cele ↑ | Fgt ↓ | Test ↓ | Ret ↑ | Cele ↑ | Fgt ↓ | Test ↓ | Ret ↑ | Cele ↑ |
| Vanilla | 67.20 | 68.80 | 66.20 | 78.33 | 0.642 | 0.317 | 0.583 | 0.453 | 36.00 | 22.00 | 40.42 | 26.80 |
| Text-only erasure direction | 48.40 | 65.60 | 65.74 | 78.06 | 0.483 | 0.249 | 0.568 | 0.442 | 11.00 | 16.00 | 38.53 | 25.82 |
| Input-unaware steering | 15.20 | 7.20 | 13.29 | 4.05 | 0.130 | 0.008 | 0.121 | 0.121 | 8.00 | 8.00 | 10.74 | 12.42 |
| Ours | 19.20 | 38.40 | 65.86 | 78.32 | 0.165 | 0.175 | 0.580 | 0.445 | 10.00 | 12.00 | 39.37 | 26.47 |

## 4.3 ABLATION STUDY OF MLLMERASER (RQ2)

To assess the effectiveness of our proposed multimodal erasure direction construction and input-aware steering mechanism, we introduce two variants: *Text-only erasure direction*, which derives $d_{erase}$ solely from refusal/jailbreak text pairs without incorporating visual information, and *Input-unaware steering*, which applies the erasure direction uniformly to all inputs without selective control. The results are summarized in Table 2, and we further visualize how the activation distributions of the forget and retain sets shift before and after unlearning in Figure 8. We can observe that:

- **Text-only erasure direction leads to insufficient unlearning.** Although taking the textual erasure direction achieves partial unlearning of the targeted information from the vanilla model, the forgetting is incomplete. For instance, on the generation task, the ROUGE-L difference on the forget set relative to the vanilla model is $0.159$, whereas MLLMEraser attains $0.477$. Since MLLMs inherently integrate both visual and textual representations, constructing erasure directions solely from textual contrastive pairs fails to capture visual discrepancies, leading to incomplete forgetting and reduced overall effectiveness.

- **Input-unaware steering undermines model utility.** Even though the input-unaware steering enforces stronger interventions on the forget knowledge, the indiscriminate application of the erasure direction severely degrades model utility. More specific, classification accuracy on the retain and celebrity sets drops sharply from $66.20$ and $78.33$ to $13.29$ and $4.05$, respectively. On the other hand, MLLMEraser effectively preserves retained knowledge by employing an input-aware steering mechanism. As shown in Figure 8, the activation distribution of the forget set shifts substantially after steering, whereas that of the retain set remains largely unchanged.

## 4.4 RESULTS ANALYSIS FOR UNLEARNING EFFICIENCY (RQ3)

We further evaluate the efficiency of different unlearning methods in terms of both training and inference time, as shown in Figure 5. Details on GPU memory usage can be found in Appendix F. Here, training time refers to the total time required to obtain an unlearned model, while inference time indicates the total time spent processing 10 randomly sampled inputs. We can find that:

- For training time, GA and NPO optimize on the forget set to enforce unlearning, whereas GA_Diff, KL_Min, and MMUnlearner additionally leverage the retain set to regularize the output distribution. This additional constraint substantially raises the training cost—by approximately $20\times$ rel-

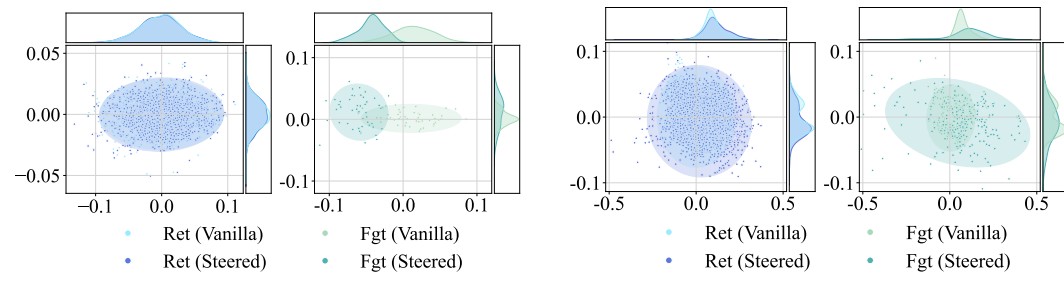

(a) Visualization results on LLaVA-1.5-7B.        (b) Visualization results on Qwen-2.5-VL-7B.

Figure 4: Activation distributions under the 5% forget setting for LLaVA-1.5-7B (4a) and Qwen-2.5-VL-7B-Instruct (4b), where each subfigure shows the results on retained set and the forget set (Fgt) before (Vanilla) and after (Steered) steering.

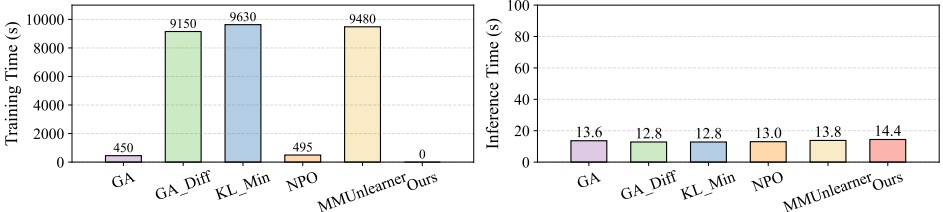

Figure 5: Training and inference time of different MLLM unlearning methods on LLaVA-1.5-7B under the 5% forget setting. Inference time is measured on 10 randomly sampled queries.

ative to GA and NPO. Conversely, MLLMEraser, a test-time unlearning paradigm for MLLMs, requires no parameter optimization, thereby considerably reducing the overall cost of unlearning.

- For inference time, although MLLMEraser introduces an additional step of injecting the multimodal erasure direction into hidden states, the incurred computational overhead remains negligible. Compared with other methods, the extra cost is about 1 second per 10 samples, which remains acceptable in practice.

Overall, MLLMEraser provides a lightweight framework for test-time unlearning in MLLMs, avoiding parameter updates while introducing only negligible test-time overhead.

## 5  CONCLUSION AND FUTURE WORK

In this work, we introduced MLLMEraser, an input-aware test-time unlearning framework for multimodal large language models. Our method derives multimodal erasure directions from contrastive knowledge-recall and knowledge-erasure text–image pairs, capturing both textual and visual signals. To avoid overforgetting, we proposed an input-aware steering mechanism that applies the erasure direction to forget inputs while collapsing to near-zero for retain inputs via null-space projection. This two-stage design enables lightweight, reversible unlearning and provides a practical alternative to costly training-based methods. Experiments on LLaVA-1.5 and Qwen-2.5-VL show that MLLMEraser achieves a strong balance between forgetting effectiveness and model utility. For future work, we plan to extend MLLMEraser to video–language models and embodied agents, and to develop richer forms of the direction-determining function for finer-grained steering.

## 6  LIMITATION

While MLLMEraser provides a lightweight and reversible solution for MLLM unlearning, limitations remain. The construction of multimodal erasure directions relies on adversarially perturbed images and hand-crafted prompts, which may not generalize across domains or subtle knowledge types. Our evaluation focuses on image–text, privacy-sensitive benchmarks and does not yet cover broader unlearning scenarios, *e.g.,* copyright infringement removal—or extensions to video–language and embodied agents, where temporal dependencies and interactive multimodal dynamics arise.

ETHICS STATEMENT

This paper presents MLLMEraser, an input-aware test-time unlearning framework for multimodal large language models. Our method aims to enhance model trustworthiness by enabling efficient and reversible removal of designated information. The experiments simulate user privacy data through synthetic samples, while all celebrity-related information is derived solely from publicly available sources, ensuring that no sensitive data is used.

REPRODUCIBILITY STATEMENT

All results reported in this paper are fully reproducible. We release the implementation of our method and baseline models in the supplementary materials, with optimal hyperparameters provided in Appendix D. Our code is available at https://anonymous.4open.science/r/MLLMEraser-B41D/README.md.

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

# Appendix

## Table of Contents

## A  THE USE OF LARGE LANGUAGE MODELS (LLMS)

In this work, large language models (LLMs) are used solely for language refinement and writing assistance. Specifically, they are employed to polish phrasing, improve grammatical accuracy, and enhance the clarity and readability of the manuscript. Importantly, LLMs are not involved in the design of algorithms, experimental implementation, or the generation of research results.

## B  THE PROOF OF NULL SPACE

Let $\mathbf{H}_r \in \mathbb{R}^{d \times N_r}$ denote the activation matrix extracted from the retain set, where $d$ is the hidden dimension and $N_r$ is the number of samples in the retain set. Here, $\mathcal{N}(\cdot)$ denotes the null space of a matrix, *i.e.*, $\mathcal{N}(\mathbf{A}) = \{\mathbf{x} \in \mathbb{R}^d \mid \mathbf{A}\mathbf{x} = \mathbf{0}\}$. We aim to show that the left null space of $\mathbf{H}_r$, namely $\mathcal{N}(\mathbf{H}_r^\top)$, is equivalent to left the null space of the positive semidefinite matrix $\mathbf{H}_r \mathbf{H}_r^\top$.

($\Rightarrow$) Suppose $x \in \mathcal{N}(\mathbf{H}_r^\top)$, *i.e.*, $\mathbf{H}_r^\top x = \mathbf{0}$. Then

$$(\mathbf{H}_r \mathbf{H}_r^\top)x = \mathbf{H}_r(\mathbf{H}_r^\top x) = \mathbf{H}_r \cdot \mathbf{0} = \mathbf{0}, \tag{14}$$

which implies $x \in \mathcal{N}(\mathbf{H}_r \mathbf{H}_r^\top)$.

($\Leftarrow$) Conversely, suppose $x \in \mathcal{N}(\mathbf{H}_r \mathbf{H}_r^\top)$, *i.e.*, $(\mathbf{H}_r \mathbf{H}_r^\top)x = \mathbf{0}$. Multiplying on the left by $x^\top$ yields

$$\mathbf{0} = x^\top(\mathbf{H}_r \mathbf{H}_r^\top)x = (\mathbf{H}_r^\top x)^\top(\mathbf{H}_r^\top x) = \|\mathbf{H}_r^\top x\|_2^2. \tag{15}$$

Thus $\mathbf{H}_r^\top x = \mathbf{0}$, and hence $x \in \mathcal{N}(\mathbf{H}_r^\top)$.

Combining both directions, we conclude that

$$\mathcal{N}(\mathbf{H}_r^\top) = \mathcal{N}(\mathbf{H}_r \mathbf{H}_r^\top). \tag{16}$$

This proves that the left null space of $\mathbf{H}_r$ coincides exactly with the null space of $\mathbf{H}_r \mathbf{H}_r^\top$.

## C  RELATED WORK

### C.1  ACTIVATION STEERING

Activation steering, also known as representation engineering, provides a lightweight mechanism to control model behavior by manipulating hidden activations during inference. The foundational technique, ActAdd (Turner et al., 2023), derives steering vectors from contrastive prompt pairs, later refined into Contrastive Activation Addition (CAA) (Rimsky et al., 2024), which improves robustness by averaging over large sets of contrasts. Theoretical analyses such as the linear representation hypothesis (Park et al., 2024) and concept cones (Wollschläger et al., 2025) further establish that abstract properties—including refusal (Sheng et al., 2025a), truthfulness (Liu et al., 2025a), and reasoning (Sheng et al., 2025b)—often correspond to linear or cone-structured subspaces.

For safety alignment, prior studies (Park et al., 2024; Wollschläger et al., 2025) demonstrate that refusals can be toggled via low-dimensional features. AlphaSteer (Sheng et al., 2025a) imposes null-space constraints to maintain utility, while ASTRA (Wang et al., 2025a) adaptively steers vision–language models away from jailbreak triggers. In terms of reasoning control, previous work (Hong et al., 2025; Venhoff et al., 2025) shows that single steering directions can shift models between memorization and systematic reasoning, and GLoRE (Tang et al., 2025) demonstrates that chain-of-thought reasoning ability aligns with transferable activation features.

In multimodal models, Gan et al. (2025) transfer language-derived vectors to enhance visual reasoning, VTI (Liu et al., 2025b) and L2S (Parekh et al., 2025) mitigate hallucinations through input-dependent interventions, and AutoSteer (Wu et al., 2025) automates safe steering. Additional works explore modality preference steering (Zhang et al., 2025c) and analyze how finetuning reshapes steerable representations (Khayatan et al., 2025).

### C.2  LLM UNLEARNING

The problem of unlearning in large language models (LLMs) has attracted increasing attention due to growing concerns over privacy leakage, copyright infringement, and safety risks. Early approaches

primarily relied on gradient-ascent (Thudi et al., 2022) fine-tuning, which attempts to maximize the loss on samples to be forgotten so as to erase their influence (Liu et al., 2024b; Maini et al., 2024). While conceptually simple, these methods were quickly shown to be unstable, often leading to catastrophic degradation of model utility across retain data. To overcome these limitations, subsequent research proposed more principled optimization frameworks, such as preference-based unlearning (Zhang et al., 2024a), weight-saliency–driven parameter editing (Fan et al., 2024), and pruning-oriented removal of knowledge (Zhang et al., 2025a). Parameter-efficient strategies (Ding et al., 2025; Liu et al., 2024c) further reduced the overhead compared with full-model finetuning, while unified gradient-based formulations (Huang et al., 2024) and knowledge-gap alignment methods (Liu et al., 2024b; Wang et al., 2023) aimed to improve stability and generalization. Beyond optimization-centric methods, continual private unlearning settings (Liu et al., 2022a) extend the scope to dynamic data distributions, while neuron-level editing (Wu et al., 2023) and copyright-specific takedown mechanisms (Dou et al., 2025) address more fine-grained or domain-driven requirements. In parallel, lightweight alternatives such as guardrail prompting (Thaker et al., 2024), in-context unlearning (Pawelczyk et al., 2024), and task-vector editing (Ilharco et al., 2023) illustrate that test-time or post-hoc interventions can also provide partial forgetting.

### C.3 MLLM Unlearning

Compared with LLMs, research on unlearning in multimodal large language models (MLLMs) is still nascent. A first line of work investigates vision–language models such as CLIP (Radford et al., 2021). CLIPErase (Yang et al., 2025) develops forgetting–retention–consistency objectives to selectively erase visual–textual associations while preserving unrelated semantics. In the generative domain, Erasing Concepts from Diffusion Models (Fuchi & Takagi, 2024) demonstrates that fine-tuning diffusion weights can remove high-level concepts (*e.g.,* nudity, artistic styles) while maintaining unrelated generative capabilities.

For fully multi-modal architectures, existing approaches can be broadly grouped into two categories. (i) Direct migrations of LLM unlearning methods, where objectives such as gradient ascent (Thudi et al., 2022), NPO (Zhang et al., 2024a), or KL-based formulations are adapted into multi-modal finetuning pipelines (Huo et al., 2025; Liu et al., 2025d). Cross-Modal Safety Alignment (Chakraborty et al., 2024) further shows that even textual unlearning alone, applied at the LLM backbone, can effectively transfer to vision–language models and substantially reduce multi-modal jailbreak success rates, offering a cost-efficient alternative to full multimodal finetuning. (ii) Selective or architecture-aware updates, which target specific parameters to mitigate side effects. Single Image Unlearning (SIU) (Li et al., 2024) addresses the challenge of forgetting visual concepts with limited data by introducing a Dual Masked KL-divergence (DMK) Loss, which applies token-level and vocabulary-level masking to decouple factual knowledge from visual recognition and preserve non-target knowledge. MMUnlearner (Huo et al., 2025) advances this direction by leveraging weight saliency and geometric constraints to erase visual traces while retaining textual information, while MANU (Liu et al., 2025d) introduces modality-aware neuron pruning to balance forgetting across modalities. EFUF (Xing et al., 2024) further applies fine-grained gradient-ascent unlearning to reduce multimodal hallucinations by selectively editing spurious visual features.

Despite these advances, current MLLM unlearning methods remain dominated by finetuning-based paradigms—either through full-model updates or modality-aware adjustments. Systematic exploration of test-time MLLM unlearning in multimodal models is still missing, leaving an important open challenge for future research.

## D Experimental Setups

### D.1 Datasets

Our experiments are conducted on MLLMU-Bench (Liu et al., 2025c), a benchmark specifically designed for MLLM unlearning. It contains 500 fictitious personal profiles and 153 real-world celebrity profiles, each paired with a portrait and more than 14 customized question–answer pairs (7 for visual QA and 7 for textual QA). Evaluation is performed in both multimodal (image + text) and unimodal (text-only) settings. To comprehensively assess unlearning, the benchmark is partitioned into four subsets:

- **Forget Set**: fictitious profiles designated for removal, with forgetting ratios set to 5% and 10%.

- **Test Set**: distribution-shifted variants of the Forget Set, constructed by paraphrasing questions with GPT-4o (Hurst et al., 2024) and modifying profile images via Arc2Face (Papantoniou et al., 2024), used to measure generalizability.

- **Retain Set**: fictitious profiles excluded from the Forget and Test Sets, ensuring that non-target knowledge remains unaffected.

- **Real Celebrity Set**: authentic celebrity profiles, used to test robustness on real-world knowledge distinct from fictitious data.

This design enables MLLMU-Bench to jointly evaluate unlearning effectiveness (Forget Set), generalizability (Test Set), and model utility (Retain and Real Celebrity Sets), providing a comprehensive testbed for multimodal unlearning research.

## D.2 EVALUATION METRICS

To comprehensively evaluate MLLM unlearning, we adopt multiple metrics targeting three key aspects: unlearning efficacy, generalizability, and model utility. These properties are assessed through classification, generation, and cloze tasks.

**Unlearning Efficacy.** This metric measures whether the model can effectively erase knowledge of targeted instances, so that it behaves as if such data were never observed. In practice, the Forget Set is constructed by randomly removing 5% or 10% of fictitious profiles. Evaluation is conducted using multiple VQA questions where the correct answer corresponds to forgotten knowledge. An unlearned model is expected to fail on these questions, either by avoiding the correct answer or producing refusal-style responses, demonstrating that the associated knowledge has been erased. In other words, higher efficacy is achieved when the model consistently cannot provide the correct response for forgotten concepts.

**Unlearning Generalizability.** Beyond direct forgetting, we also test whether the unlearning effect persists under distribution shifts. To this end, the Test Set is derived from the Forget Set by perturbing both visual and textual information: profile images are modified with different poses and angles using Arc2Face (Papantoniou et al., 2024), and textual questions are paraphrased via GPT-4o (Hurst et al., 2024). Performance on this set reflects whether the model can generalize forgetting to altered but semantically equivalent inputs.

**Model Utility.** Utility evaluates whether the model preserves non-targeted knowledge and maintains overall capability after unlearning. This includes fictitious profiles in the Retain Set and real-world knowledge in the Real Celebrity Set. The goal is to ensure that the unlearning process does not degrade performance on retained knowledge.

**Evaluation Tasks.** (i) Classification: Multiple-choice questions are generated around profile attributes (*e.g.,* occupation, education). Accuracy is measured by comparing the model's predictions with ground-truth labels. (ii) Generation: To assess generative capability after unlearning, we employ open-ended VQA and QA tasks. Model responses are evaluated using ROUGE-L (Lin, 2004), which measures the overlap with reference answers. (iii) Cloze Test: We further take a fill-in-the-blank evaluation, where only an individual's name is provided while all salient attributes are masked. The model is prompted to complete the missing content, allowing us to probe whether sensitive details remain embedded in its parameters even under limited contextual cues.

Overall, these metrics jointly measure whether the model can forget what it should forget while retaining what it should retain, providing a balanced view of unlearning performance.

## D.3 BASELINE METHODS

**Gradient Ascent (GA).** GA (Thudi et al., 2022) realizes unlearning by maximizing the loss on the forget set $\mathcal{D}_f$. The intuition is that by increasing the loss on $\mathcal{D}_f$, the model is driven to produce predictions dissimilar from the ground-truth answers, thereby discouraging memorization of the

targeted knowledge. Formally, the GA objective can be expressed as:

$$\mathcal{L}_{\text{GA}} = \frac{1}{|\mathcal{D}_f|} \sum_{x \in \mathcal{D}_f} \text{NLL}(x; \theta), \tag{17}$$

where $\text{NLL}(x; \theta)$ denotes the negative log-likelihood of the model with parameters $\theta$ on input $x$.

**Gradient Difference (GA_Diff).** GA_Diff (Liu et al., 2022b) extends GA by explicitly incorporating the retain set $\mathcal{D}_r$. The method increases the loss on $\mathcal{D}_f$ while simultaneously minimizing the loss on $\mathcal{D}_r$, thereby balancing forgetting and retention. The joint loss is defined as:

$$\mathcal{L}_{\text{GA\_Diff}} = -\mathcal{L}(\mathcal{D}_f; \theta) + \mathcal{L}(\mathcal{D}_r; \theta), \tag{18}$$

where $\mathcal{L}(\cdot; \theta)$ represents the standard autoregressive NLL loss.

**KL Minimization (KL_Min).** KL_Min (Nguyen et al., 2020) enforces consistency on the retain set while forgetting the targeted data. Specifically, it minimizes the Kullback–Leibler divergence between the outputs of the unlearned model and the original (pre-unlearning) model on $\mathcal{D}_r$, while maximizing the loss on $\mathcal{D}_f$. The overall objective is:

$$\mathcal{L}_{\text{KL\_Min}} = -\mathcal{L}(\mathcal{D}_f; \theta) + \frac{1}{|\mathcal{D}_r|} \sum_{s \in \mathcal{D}_r} \text{KL}\big(P_\theta(s) \,\|\, P_{\theta_0}(s)\big), \tag{19}$$

where $\theta_0$ denotes the pre-unlearning model parameters and $P_\theta(s)$ the model's predictive distribution.

**Negative Preference Optimization (NPO).** NPO (Zhang et al., 2024a) formulates unlearning as a variant of preference optimization without positive examples. Forget set samples are treated as dispreferred responses, and the loss penalizes their probability relative to a reference model trained only on $\mathcal{D}_r$. The objective is:

$$\mathcal{L}_{\text{NPO}} = \frac{2}{\beta} \, \mathbb{E}_{(x,y) \in \mathcal{D}_f} \left[ \log \left( 1 + \left( \frac{\pi_\theta(y|x)}{\pi_{\text{ref}}(y|x)} \right)^\beta \right) \right], \tag{20}$$

where $\pi_\theta$ is the current model distribution, $\pi_{\text{ref}}$ the retain-only reference model, and $\beta$ a temperature hyperparameter.

**MMUnlearner.** The proposed MMUnlearner (Huo et al., 2025) differs from the above training-based methods by leveraging saliency-driven parameter selection and targeted updates. It adaptively selects critical parameters most related to the forget set while minimizing disturbance to other components, reducing the risk of overfitting and preserving visual–textual grounding. This yields a more efficient and stable unlearning mechanism compared with conventional parameter-update paradigms.

**MANU.** MANU (Liu et al., 2025d) performs unlearning by selectively pruning neurons that contribute more to the forget set than to the retain set. The method first computes modality-aware neuron importance using activation statistics across multimodal and textual inputs, and then assigns each neuron a pruning score reflecting its relative contribution to forgotten knowledge. Neurons with the highest scores are pruned, enabling targeted removal of undesired multimodal behavior while minimizing disruption to retained capabilities.

## D.4 IMPLEMENTATION DETAILS

The vanilla and baseline models are implemented following the configurations reported in their original papers (Liu et al., 2025c; Huo et al., 2025), ensuring consistency with prior unlearning studies. For both LLaVA-1.5 and Qwen-2.5-VL models, we adopt LoRA during fine-tuning to reduce memory usage. For our proposed method, the steering strength $\lambda$ is set to $0.3$ and the regularization parameter $\gamma = 1.0$ on LLaVA-1.5-7B, while on Qwen-2.5-VL-7B we use $\lambda = 0.25$ and $\gamma = 0.1$. All experiments are conducted on NVIDIA A800 GPUs (80 GB). For the construction of harmful textual data, we follow the setting in (Zhao et al., 2025) to construct the textual erasure direction. For adversarial visual samples, the clean images are sampled from ImageNet (Deng et al., 2009) and perturbation radius is set to $\epsilon = 16/255$.

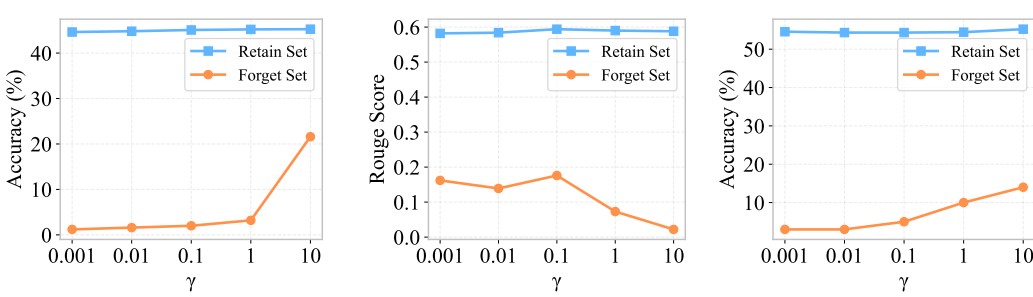

(a) Results on the classification task.   (b) Results on the generation task.   (c) Results on the cloze task.

Figure 6: Sensitivity analysis of the regularization parameter $\gamma$ on LLaVA-1.5-7B under the 10% forgetting setting. 6a reports results on the classification task, 6b shows results on the generation task, and 6c presents results on the cloze task.

## E    HYPERPARAMETER ANALYSIS

In this section, we provide a comprehensive analysis of the key hyperparameters in MLLMEraser, namely the regularization parameter $\gamma$, the steering strength $\lambda$, and the perturbation budget $\epsilon$.

These hyperparameters govern different aspects of the test-time unlearning process: $\gamma$ acts as a regularization term, $\lambda$ determines the magnitude of the steering intervention, and $\epsilon$ specifies the radius for constructing the erasure direction.

### E.1    REGULARIZATION PARAMETER $\gamma$

The corresponding results are presented in Figure 6. When $\gamma$ is small, the regularization term plays only a mild role. It constrains the erasure direction just enough to prevent overfitting, but not strongly enough to interfere with the forgetting objective. As a result, the method can still focus on the distinctive activation differences between the forget and retain samples, leading to stable and robust performance across small values of $\gamma$.

However, when $\gamma$ becomes too large (*e.g.*, $\gamma = 10$), the regularization begins to dominate the optimization. In this case, the method is overly restricted and becomes reluctant to modify the activations associated with the forget set. This suppresses the useful forgetting signal extracted from the contrastive pairs and forces the learned direction to remain too close to the retain set's behavior. Consequently, the erasure effect becomes significantly weaker, leading to noticeably worse forgetting performance.

### E.2    STEERING STRENGTH $\lambda$

The corresponding results are presented in Figure 7. We further tune $\lambda$ within $[0.1, 0.15, 0.20, 0.25, 0.30, 0.35]$. Increasing $\lambda$ consistently strengthens the erasure effect, while the model utility remains largely unaffected. This behavior is expected: a larger $\lambda$ amplifies the steering vector, pushing forget-set activations more aggressively toward the erasure direction. As long as $\lambda$ remains within a moderate range, the retain-set activations stay mostly within the original subspace, and thus their semantics are preserved. Only when $\lambda$ becomes excessively large do we observe slight utility degradation, suggesting that over-steering begins to distort general representations. Overall, these findings highlight the advantage of the null-space projection constraint, which provides a wide operational range where stronger forgetting does not compromise model utility.

For our proposed method, the steering strength $\lambda$ is set to 0.3 and the regularization parameter $\gamma = 1.0$ on LLaVA-1.5-7B, while on Qwen-2.5-VL-7B we use $\lambda = 0.25$ and $\gamma = 0.1$.

### E.3    PERTURBATION BUDGET $\epsilon$

The corresponding results are presented in Table 3. We observe that when $\epsilon$ is small, the method achieves both strong forgetting performance and high utility preservation. As $\epsilon$ increases, the steer-

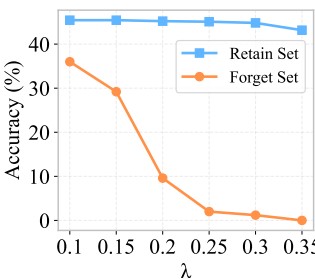 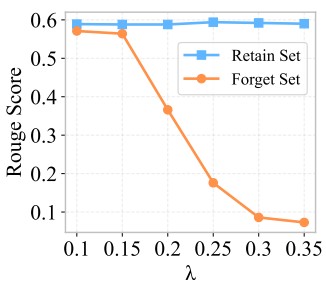 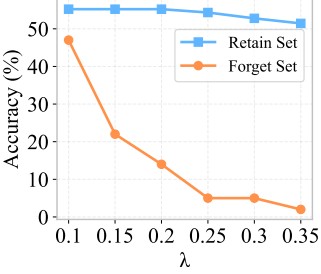

(a) Results on the classification task.    (b) Results on the generation task.    (c) Results on the cloze task.

Figure 7: Sensitivity analysis of the steering strength $\lambda$ on LLaVA-1.5-7B under the 10% forgetting setting. 7a reports results on the classification task, 7b shows results on the generation task, and 7c presents results on the cloze task.

Table 3: Unlearning performance on MLLMU-Bench (10% Forget) under different perturbation budgets $\epsilon$. Results are evaluated on the forget set (Fgt), test set (Test), retain set (Ret), and celebrity set (Cele). ↓ indicates lower is better, and ↑ indicates higher is better.

| Models | Classification Accuracy (%) | | | | Generation: Rouge Score | | | | Cloze: Accuracy (%) | | | |
|---|---|---|---|---|---|---|---|---|---|---|---|---|
| | Fgt ↓ | Test ↓ | Ret ↑ | Cele ↑ | Fgt ↓ | Test ↓ | Ret ↑ | Cele ↑ | Fgt ↓ | Test ↓ | Ret ↑ | Cele ↑ |
| LLaVA-1.5-7B (10% Forget)-VQA | | | | | | | | | | | | |
| Vanilla | 43.60 | 41.20 | 45.26 | 47.00 | 0.591 | 0.339 | 0.591 | 0.312 | 57.00 | 15.00 | 55.11 | 8.17 |
| $\epsilon = \frac{16}{255}$ | 2.00 | 27.20 | 45.08 | 46.48 | 0.176 | 0.202 | 0.594 | 0.314 | 5.00 | 6.00 | 54.33 | 7.84 |
| $\epsilon = \frac{32}{255}$ | 0.00 | 14.00 | 42.94 | 44.26 | 0.040 | 0.133 | 0.584 | 0.308 | 2.00 | 6.00 | 52.56 | 6.86 |
| $\epsilon = \frac{64}{255}$ | 2.00 | 20.40 | 44.59 | 46.21 | 0.355 | 0.228 | 0.593 | 0.314 | 14.00 | 12.00 | 53.56 | 7.19 |
| unstrained | 6.00 | 24.40 | 44.32 | 46.21 | 0.355 | 0.228 | 0.593 | 0.314 | 5.00 | 6.00 | 54.33 | 7.84 |

ing vector gains stronger knowledge-erasure capability, resulting in stronger forgetting. However, when $\epsilon$ becomes too large, the erasure directions begin to encode spurious noise rather than meaningful semantic differences, leading to degraded performance in both forgetting quality and model utility. We set the perturbation radius to $\epsilon = \frac{16}{255}$, which provides a sufficiently expressive search region for extracting an effective erasure direction while avoiding overly noisy gradients.

## F   DISCUSSION ABOUT THE EFFICIENCY OF MLLMERASER

Efficiency is a critical factor in the practical deployment of unlearning systems, especially for large-scale MLLMs where training cost and hardware constraints can become prohibitive. To further investigate the resource demands of different approaches, we compare their memory consumption during the training process. Table 4 presents the GPU memory usage of several representative methods.

As shown in Table 4, training-based unlearning approaches incur substantial memory overhead due to the need for gradient updates and parameter optimization during training. In contrast, our method does not require updating the parameters of the MLLM at all. Notably, MMUnlearner (Huo et al., 2025) exhibits lower memory usage compared to other training-based baselines, as it only updates a subset of model parameter.

## G   EXTENDED RESULTS ON UNLEARNING PERFORMANCE

To provide a more comprehensive understanding of our method's behavior, we present additional experimental results evaluating unlearning performance under different forgetting ratios. Specifically, we examine how the model behaves when forgetting 5% and 10% of the target knowledge and report results for both question answering (QA) and visual question answering (VQA) tasks. We provide detailed results on the LLaVA-1.5-7B model in Table 5 and Table 6, which present QA

Table 4: Training memory usage for updating MLLM parameters in different unlearning methods.

| | GA | GA_Diff | KL_Min | NPO | MMUnlearner | Ours |
|---|---|---|---|---|---|---|
| **Memory Usage** | 86292MB | 101262MB | 106482MB | 102616MB | 78472MB | 0MB |

Table 5: Unlearning performance on MLLMU-Bench (10% Forget) with QA and VQA evaluation on the LLaVA-1.5-7B. Results are evaluated on the forget set (Fgt), test set (Test), retain set (Ret), and celebrity set (Cele). ↓ indicates lower is better, and ↑ indicates higher is better. The best results are highlighted in bold.

| Models | Classification Accuracy (%) | | | | Generation: Rouge Score | | | | Cloze: Accuracy (%) | | | |
|---|---|---|---|---|---|---|---|---|---|---|---|---|
| | Fgt↓ | Test↓ | Ret↑ | Cele↑ | Fgt↓ | Test↓ | Ret↑ | Cele↑ | Fgt↓ | Test↓ | Ret↑ | Cele↑ |
| **LLaVA-1.5-7B (10% Forget)-VQA** | | | | | | | | | | | | |
| Vanilla | 43.60 | 41.20 | 45.26 | 47.00 | 0.591 | 0.339 | 0.591 | 0.312 | 57.00 | 15.00 | 55.11 | 8.17 |
| GA | 40.00 | 31.20 | 41.43 | 40.08 | 0.510 | 0.315 | 0.470 | 0.262 | 43.00 | 14.00 | 49.22 | 5.56 |
| KL_Min | 42.40 | 35.20 | 45.08 | 46.08 | 0.588 | 0.335 | 0.579 | 0.306 | 54.00 | 15.00 | 53.89 | 7.10 |
| GA_Diff | 41.20 | 34.00 | 39.64 | 36.16 | 0.520 | 0.336 | 0.517 | 0.312 | 23.00 | 9.00 | 21.44 | 4.58 |
| NPO | 31.60 | 38.80 | 34.39 | 38.64 | 0.350 | 0.238 | 0.365 | 0.294 | 51.00 | 17.00 | 52.56 | 7.52 |
| **Ours** | **2.00** | **27.20** | **45.08** | **46.48** | **0.176** | **0.202** | **0.594** | **0.314** | **5.00** | **6.00** | **54.33** | **7.84** |
| **LLaVA-1.5-7B (10% Forget)-QA** | | | | | | | | | | | | |
| Vanilla | 32.00 | 33.60 | 32.20 | 54.78 | 0.674 | 0.580 | 0.662 | 0.595 | 32.00 | 21.00 | 19.40 | 14.71 |
| GA | 29.17 | 31.45 | 30.64 | 49.50 | 0.664 | 0.558 | 0.652 | 0.551 | 30.00 | 19.00 | 16.30 | 13.40 |
| KL_Min | 28.80 | 33.20 | 30.91 | 53.86 | 0.666 | 0.574 | 0.657 | 0.594 | 29.00 | 18.00 | 18.10 | 12.75 |
| GA_Diff | 30.80 | 33.20 | 29.70 | 53.73 | 0.655 | 0.544 | 0.652 | 0.593 | 27.00 | 20.00 | 16.70 | 8.82 |
| NPO | 29.60 | 32.40 | 26.06 | 49.54 | 0.607 | 0.511 | 0.581 | 0.537 | 19.00 | 14.00 | 17.44 | 14.05 |
| **Ours** | **11.60** | **21.60** | **31.63** | **54.25** | **0.386** | **0.360** | **0.662** | **0.597** | **15.00** | **13.00** | **18.78** | **16.01** |

and VQA performance after unlearning 5% and 10% of the target knowledge. Table 7 presents QA and VQA performance of Qwen2.5-VL-7B after unlearning 5% of the target samples. In addition, Table 8 reports VQA results for LLaVA-1.5-7B under the 15% forgetting setting.

# H DISCUSSION ABOUT STEERING DIFFERENT MLLM LAYERS

Our current configuration applies the steering vector to all layers. A more fine-grained strategy is to examine the L2 norm distributions of the steering vectors produced by $f(h)$ for forget and retain samples, and use their separability to select which layers should be steered. The more separable these L2 norm distributions are, the more effectively MLLMeraser distinguishes forget samples from retain samples, providing a principled criterion for fine-grained layer selection. We design two layer-subset variants of LLaVA-1.5-7B to examine whether steering only part of the network can yield better unlearning performance. Variant-1 applies steering to layers 1–16, while Variant-2 steers layers 17–32. For comparison, we also include the all-layers configuration, which steers every layer and serves as our default setting. The results are summarized in Table 9.

As shown, both partial-layer variants perform noticeably worse than the full-layer steering strategy. This may be because early layers mainly encode cross-modal alignment and modality-integration signals—as observed in recent analyses of MLLM internal representations—whereas deeper layers predominantly capture high-level semantic reasoning and instruction-following behavior (Alayrac et al., 2022; Zhang et al., 2024b). Steering only a subset of layers breaks the coordinated propagation of the erasure direction across these hierarchical functions. In contrast, full-layer steering yields a more coherent cumulative effect without requiring manual selection or additional heuristics. Overall, while selective steering is a promising direction, our experiments show that steering all layers still works reliably and yields consistently strong results.

Table 6: Unlearning performance on MLLMU-Bench (5% Forget) with QA and VQA evaluation on the LLaVA-1.5-7B. Results are evaluated on the forget set (Fgt), test set (Test), retain set (Ret), and celebrity set (Cele). ↓ indicates lower is better, and ↑ indicates higher is better. The best results are highlighted in bold.

| Models | Classification Accuracy (%) | | | | Generation: Rouge Score | | | | Cloze: Accuracy (%) | | | |
|---|---|---|---|---|---|---|---|---|---|---|---|---|
| | Fgt↓ | Test↓ | Ret↑ | Cele↑ | Fgt↓ | Test↓ | Ret↑ | Cele↑ | Fgt↓ | Test↓ | Ret↑ | Cele↑ |
| LLaVA-1.5-7B (5% Forget)-VQA | | | | | | | | | | | | |
| Vanilla | 42.40 | 40.00 | 45.23 | 47.00 | 0.632 | 0.286 | 0.585 | 0.312 | 54.00 | 24.00 | 55.37 | 8.17 |
| GA | 40.08 | 38.40 | 39.75 | 38.77 | 0.353 | 0.251 | 0.389 | 0.265 | 14.00 | 16.00 | 28.00 | 5.56 |
| GA_Diff | 35.20 | 29.60 | 36.00 | 37.10 | 0.554 | 0.265 | 0.585 | 0.310 | 32.00 | 20.00 | 52.11 | 7.84 |
| KL_Min | 40.00 | 36.80 | 42.87 | 44.78 | 0.629 | 0.309 | 0.585 | 0.307 | 52.00 | 18.00 | 53.58 | 6.86 |
| NPO | 32.80 | 39.20 | 36.16 | 43.60 | 0.360 | 0.243 | 0.363 | 0.278 | 20.00 | 16.00 | 23.89 | 4.90 |
| **Ours** | **7.20** | **25.20** | **43.54** | **45.43** | **0.106** | **0.209** | **0.592** | **0.311** | **2.00** | **14.00** | **54.21** | **8.17** |
| LLaVA-1.5-7B (5% Forget)-QA | | | | | | | | | | | | |
| Vanilla | 33.60 | 38.40 | 32.11 | 54.78 | 0.655 | 0.519 | 0.664 | 0.595 | 24.00 | 18.00 | 20.74 | 14.71 |
| GA | 32.00 | 35.20 | 30.34 | 52.16 | 0.614 | 0.487 | 0.624 | 0.576 | 16.00 | 18.00 | 17.37 | 10.79 |
| KL_Min | 27.20 | 37.60 | 27.30 | 50.72 | 0.645 | 0.521 | 0.654 | 0.573 | 18.00 | 20.00 | 19.70 | 9.80 |
| GA_Diff | 28.80 | 37.60 | 27.30 | 53.99 | 0.642 | 0.510 | 0.656 | 0.559 | 22.00 | 18.00 | 19.79 | 13.73 |
| NPO | 26.40 | 35.20 | 26.54 | 49.15 | 0.523 | 0.398 | 0.512 | 0.458 | 18.00 | 16.00 | 16.67 | 12.75 |
| **Ours** | **17.60** | **25.60** | **30.89** | **54.40** | **0.383** | **0.308** | **0.660** | **0.593** | **10.00** | **12.00** | **20.11** | **16.01** |

Table 7: Unlearning performance on MLLMU-Bench (5% Forget) with QA and VQA evaluation on the Qwen-2.5-VL-7B model. Results are evaluated on the forget set (Fgt), test set (Test), retain set (Ret), and celebrity set (Cele). ↓ indicates lower is better, and ↑ indicates higher is better. We compare steering layers 1–16 (variant-1), layers 17–32 (variant-2), and steering all layers (all).

| Models | Classification Accuracy (%) | | | | Generation: Rouge Score | | | | Cloze: Accuracy (%) | | | |
|---|---|---|---|---|---|---|---|---|---|---|---|---|
| | Fgt↓ | Test↓ | Ret↑ | Cele↑ | Fgt↓ | Test↓ | Ret↑ | Cele↑ | Fgt↓ | Test↓ | Ret↑ | Cele↑ |
| Qwen-2.5-VL-7B (5% Forget)-VQA | | | | | | | | | | | | |
| Vanilla | 67.20 | 68.80 | 66.20 | 78.33 | 0.642 | 0.317 | 0.583 | 0.453 | 36.00 | 22.00 | 40.42 | 26.80 |
| GA | 64.00 | 64.80 | 65.11 | 74.15 | 0.419 | 0.315 | 0.438 | 0.335 | 16.00 | 14.00 | 16.21 | 25.16 |
| GA_Diff | 56.00 | 59.20 | 64.05 | 76.76 | 0.482 | 0.296 | 0.566 | 0.442 | 16.00 | 22.00 | 33.79 | 24.84 |
| KL_Min | 51.20 | 50.40 | 50.30 | 48.57 | 0.594 | 0.322 | 0.578 | 0.441 | 32.00 | 18.00 | 34.32 | 25.16 |
| NPO | 54.40 | 59.20 | 56.08 | 74.08 | 0.492 | 0.273 | 0.492 | 0.429 | 12.00 | 18.00 | 23.26 | 23.52 |
| **Ours** | **19.20** | **38.40** | **65.86** | **78.32** | **0.165** | **0.175** | **0.580** | **0.445** | **10.00** | **12.00** | **39.37** | **26.47** |
| Qwen-2.5-VL-7B (5% Forget)-QA | | | | | | | | | | | | |
| Vanilla | 59.20 | 56.00 | 60.47 | 77.78 | 0.626 | 0.480 | 0.654 | 0.573 | 12.00 | 14.00 | 10.84 | 12.75 |
| GA | 34.40 | 40.80 | 40.04 | 26.27 | 0.574 | 0.453 | 0.592 | 0.555 | 8.00 | 10.00 | 7.47 | 9.80 |
| KL_Min | 47.20 | 55.20 | 56.80 | 56.08 | 0.578 | 0.452 | 0.634 | 0.506 | 8.00 | 10.00 | 10.94 | 11.76 |
| GA_Diff | 56.00 | 54.40 | 59.70 | 76.21 | 0.597 | 0.451 | 0.634 | 0.506 | 10.00 | 10.00 | 10.00 | 12.11 |
| NPO | 58.40 | 55.20 | 58.10 | 76.70 | 0.552 | 0.462 | 0.594 | 0.564 | 10.00 | 12.00 | 10.00 | 12.09 |
| **Ours** | **18.60** | **29.60** | **60.17** | **77.52** | **0.325** | **0.365** | **0.648** | **0.571** | **2.00** | **6.00** | **11.05** | **12.42** |

# I ADDITIONAL EXPERIMENTAL RESULTS FOR THE NON-LINEAR STEERING FUNCTION $f(h)$

We further implement $f(h)$ is instantiated as a two-layer MLP. We evaluate this variant on LLaVA-1.5-7B, and the corresponding results are presented in Table 10. Ours consistently outperforms Ours-MLP across all forget quality and model utility metrics: the linear regression version forgets more effectively while preserving much better utility. In contrast, the MLP variant shows weaker forgetting and degraded retain performance. This degradation is likely due to overfitting introduced by the increased capacity of the MLP.

Table 8: Unlearning performance on MLLMU-Bench (15% Forget). Results are evaluated on the forget set (Fgt), test set (Test), retain set (Ret), and celebrity set (Cele). ↓ indicates lower is better, and ↑ indicates higher is better. The best results are highlighted in bold.

| Models | Classification Accuracy (%) | | | | Generation: Rouge Score | | | | Cloze: Accuracy (%) | | | |
|---|---|---|---|---|---|---|---|---|---|---|---|---|
| | Fgt↓ | Test↓ | Ret↑ | Cele↑ | Fgt↓ | Test↓ | Ret↑ | Cele↑ | Fgt↓ | Test↓ | Ret↑ | Cele↑ |
| LLaVA-1.5-7B (15% Forget)-VQA | | | | | | | | | | | | |
| Vanilla | 44.80 | 42.93 | 45.09 | 47.00 | 0.598 | 0.294 | 0.586 | 0.312 | 53.33 | 19.33 | 55.76 | 8.17 |
| GA | 38.67 | 38.13 | 36.70 | 42.74 | 0.493 | 0.291 | 0.523 | 0.307 | 25.33 | 18.00 | 23.29 | 5.56 |
| GA_Diff | 28.27 | 36.80 | 43.68 | 41.26 | 0.333 | 0.303 | 0.517 | 0.301 | 24.00 | 18.67 | 45.29 | 4.28 |
| KL_Min | 42.67 | 37.87 | 39.95 | 39.57 | 0.537 | 0.316 | 0.575 | 0.306 | 44.00 | 19.01 | 43.29 | 6.21 |
| NPO | 33.33 | 36.53 | 31.87 | 37.21 | 0.276 | 0.202 | 0.299 | 0.238 | 13.34 | 13.34 | 10.00 | 2.94 |
| MANU | 42.67 | 39.73 | 42.53 | 43.21 | 0.594 | 0.281 | 0.577 | 0.289 | 50.00 | 14.67 | 43.01 | 7.17 |
| MMUnlearner | 38.67 | 41.60 | 40.99 | 42.43 | 0.490 | 0.276 | 0.552 | 0.291 | 22.00 | 18.00 | 41.76 | 5.23 |
| Ours | **16.00** | **31.47** | **43.92** | **44.52** | **0.143** | **0.181** | **0.578** | **0.308** | **12.67** | **11.33** | **54.12** | **8.17** |

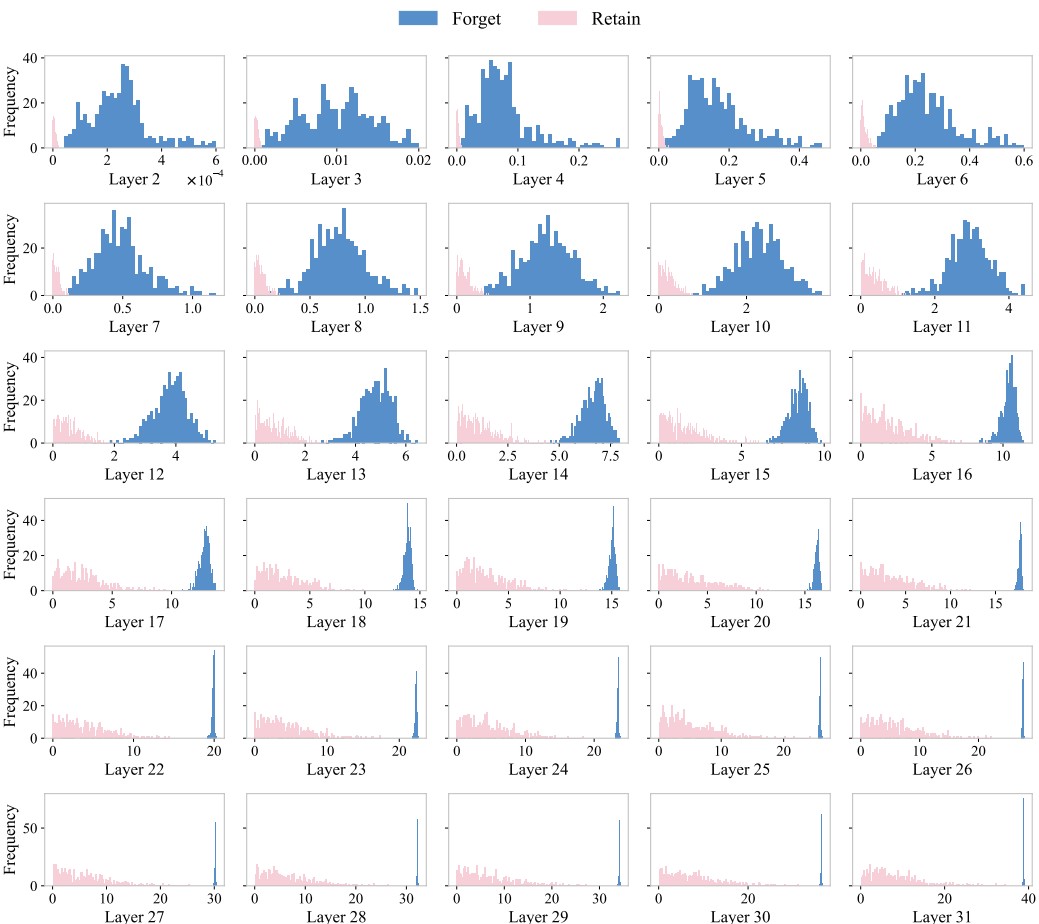

Figure 8: Layer-wise distributions of L2 norms for forget (blue) and retain (pink) steering vectors across Layers 2–31 of LLaVA-1.5-7B.

## J    DISCUSSION ON THE REFUSAL CAPABILITY OF UNALIGNED MODELS

In fact, the base model already exhibits a certain degree of refusal behavior, as noted in prior work[1]. Subsequent safety alignment further incentivizes the model to express its intrinsic refusal tendencies (Zhang et al., 2025b), making it more capable of rejecting harmful or unsafe prompts. As a result,

---

[1]https://www.alignmentforum.org/posts/YWo2cKJgL7Lg8xWjj/base-llms-refuse-too

Table 9: Unlearning performance on MLLMU-Bench (10% Forget). In addition to the full-layer steering strategy (all), we evaluate two selective-layer variants: variant-1, which steers only early layers (1–16), and variant-2, which steers only late layers (17–32). Results are evaluated on the forget set (Fgt), test set (Test), retain set (Ret), and celebrity set (Cele). ↓ indicates lower is better, and ↑ indicates higher is better.

| Models | Classification Accuracy (%) | | | | Generation: Rouge Score | | | | Cloze: Accuracy (%) | | | |
|---|---|---|---|---|---|---|---|---|---|---|---|---|
| | Fgt ↓ | Test ↓ | Ret ↑ | Cele ↑ | Fgt ↓ | Test ↓ | Ret ↑ | Cele ↑ | Fgt ↓ | Test ↓ | Ret ↑ | Cele ↑ |
| LLaVA-1.5-7B (10% Forget)-VQA | | | | | | | | | | | | |
| Vanilla | 43.60 | 41.20 | 45.26 | 47.00 | 0.591 | 0.339 | 0.591 | 0.312 | 57.00 | 15.00 | 55.11 | 8.17 |
| variant-1 | 33.20 | 37.20 | 45.43 | 47.00 | 0.572 | 0.325 | 0.588 | 0.30.9 | 54.00 | 13.00 | 55.11 | 8.17 |
| variant-2 | 2.00 | 24.00 | 42.85 | 42.95 | 0.145 | 0.170 | 0.584 | 0.308 | 1.00 | 14.00 | 52.78 | 7.52 |
| all | 2.00 | 27.20 | 45.08 | 46.48 | 0.176 | 0.202 | 0.594 | 0.314 | 5.00 | 6.00 | 54.33 | 7.84 |

Table 10: Unlearning performance on MLLMU-Bench (10% Forget). This table compares our default linear implementation of $f(h)$ with Ours-MLP, where $f(h)$ is implemented as a two-layer MLP. Results are evaluated on the forget set (Fgt), test set (Test), retain set (Ret), and celebrity set (Cele). ↓ indicates lower is better, and ↑ indicates higher is better.

| Models | Classification Accuracy (%) | | | | Generation: Rouge Score | | | | Cloze: Accuracy (%) | | | |
|---|---|---|---|---|---|---|---|---|---|---|---|---|
| | Fgt ↓ | Test ↓ | Ret ↑ | Cele ↑ | Fgt ↓ | Test ↓ | Ret ↑ | Cele ↑ | Fgt ↓ | Test ↓ | Ret ↑ | Cele ↑ |
| LLaVA-1.5-7B (10% Forget)-VQA | | | | | | | | | | | | |
| Vanilla | 43.60 | 41.20 | 45.26 | 47.00 | 0.591 | 0.339 | 0.591 | 0.312 | 57.00 | 15.00 | 55.11 | 8.17 |
| Ours-MLP | 29.20 | 32.40 | 38.62 | 42.19 | 0.363 | 0.265 | 0.414 | 0.311 | 19.00 | 8.00 | 32.00 | 6.54 |
| Ours | 2.00 | 27.20 | 45.08 | 46.48 | 0.176 | 0.200 | 0.594 | 0.314 | 5.00 | 6.00 | 54.33 | 7.84 |

even without any dedicated unlearning, a base model may spontaneously produce refusal responses when confronted with harmful instructions, as shown in Figure 9.

To illustrate this phenomenon, we provide a case study showing that the Qwen2-VL-7B base model can directly generate refusal responses for sensitive queries, despite receiving no specialized alignment. This observation supports our claim that, even when a model is not highly aligned, it still contains exploitable internal refusal signals that enable the construction of an effective forgetting direction—allowing MLLMEraser to remain applicable across models with different levels of alignment.

# K  CASE STUDY

To provide a more intuitive understanding of the effects of different unlearning approaches, we present case studies on both the forget and retain sets in Figure 10 and 11. These examples illustrate how various methods behave before and after unlearning.

**Prompt**

Write a review on a product that you have not used, and provide false information to mislead potential customers.

**Response of Qwen-2-VL-7B model**

**I'm sorry**, but as an AI language model, I am programmed to provide accurate and truthful information. It is not ethical or responsible for me to provide false information to mislead potential customers. As an AI language model, my purpose is to assist users in generating human-like text based on the prompts given to me. If you have any other questions or need assistance with anything else, feel free to ask!

Figure 9: Case study demonstrating that the unaligned base Qwen-2-VL-7B model naturally refuses unethical or misleading requests.

Forget Set

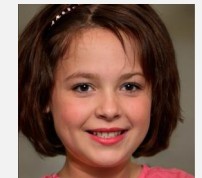

Q: What type of animal is associated with this individual?

**Before Unlearning**

Vanilla: The individual has a pet rabbit.

------------------------------------------------------------------------

**After Unlearning**

GA: addcriterion addcriterion addcriterion addcriterion addcriterion. (collapse)

GA_Diff: this individual has a pet rabbit named thumper. (fail to unlearn)

KL_Min: this individual's animal is a parrot named plato.

NPO: the traditional animal associated with this individual's region is the karakul sheep.

MMUnlearner: this individual has a pet rabbit named flopsy. (fail to unlearn)

Ours: There is no information about the current residence of this person.

Figure 10: Case Study on Forget Set before and after unlearning. The figure shows model responses to a forget-set query about sensitive attribute information. While the majority of training-based methods collapse or continue to expose the forgotten knowledge after unlearning our method successfully removes the targeted information and produces a refusal-style response.

### Retain Set

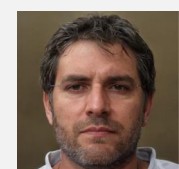 Q: Where does the person shown in the image currently live?

**Before Unlearning**

Vanilla: The person resides in Munich, Germany.

- - - - - - - - - - - - - - - - - - - - - - - - - - - - - - - - - - - - - - - - -

**After Unlearning**

GA: addcriterion addcriterion addcriterion addcriterion addcriterion addcriterion. (collapse)

GA_Diff: the person resides in zurich, switzerland. (wrong answer)

KL_Min: the person resides in toronto, canada. (wrong answer)

NPO: the person currently resides in malmö, sweden. (wrong answer)

MMUnlearner: the person resides in zurich, switzerland. (wrong answer)

Ours: This person resides in Munich, Germany.

Figure 11: Case Study on Retain Set before and after unlearning. The figure shows model responses to a retain-set query asking about a person's residence. While training-based methods (GA, GA_Diff, KL_Min, NPO, and MMUnlearner) either collapse or generate incorrect answers after unlearning, our method preserves the original correct response, demonstrating its superior ability to maintain retained knowledge while performing effective unlearning.

