# OpenReview forum: "MLLMEraser: Achieving Test-Time Unlearning in Multimodal Large Language Models through Activation Steering"
_ICLR.cc/2026/Conference — Submitted to ICLR 2026_

### Official Review · Reviewer_ssTS · 2025-10-15

**Soundness:** 3
**Presentation:** 3
**Contribution:** 3
**Rating:** 4
**Confidence:** 4

**Summary:**

MLLMEraser introduces the first test-time unlearning method for multimodal large language models (MLLMs), eliminating the need for parameter updates. Instead of retraining, it uses activation steering to dynamically erase designated knowledge during inference.

**Strengths:**

- **Efficient Algorithm:** The work redefines unlearning as a reversible inference-time process rather than a retraining problem, marking an important conceptual shift. By steering activations instead of modifying weights, MLLMEraser sidesteps gradient conflicts between forget and retain sets that typically degrade model utility.
- **Reversibility and plug-and-play design:** MLLMEraser does not modify parameters, it can be detached or re-applied instantly. This reversible property makes it suitable for real-world applications such as dynamic privacy control or content moderation, where models must flexibly enforce or lift restrictions. Few prior unlearning methods offer such operational simplicity.

**Weaknesses:**

- The direction of determining function f(h) is modeled as a linear transformation, which assumes that forget and retain samples are linearly separable in activation space, but knowledge entanglement in MLLMs can be highly non-linear, especially across visual and textual dimensions.
- MLLMU-Bench seems to have three forgetting setups (5%, 10%, 15%), and it seems like 15% case is missing across the entire paper.
- MLLMEraser assumes that the underlying MLLM can already exhibit consistent refusal behavior, which is not always the case, especially for weaker or open-source base models without robust safety tuning. If the model cannot reliably produce refusal responses, the contrastive setup for constructing the erasure direction becomes ill-posed, potentially leading to ineffective or unstable unlearning.
- The adversarial optimization used to simulate harmful visual recall (via PGD updates) introduces potential artifacts. While it successfully amplifies harmful responses, these perturbations may push the images outside the distribution of realistic inputs, leading to erasure directions that encode spurious noise rather than semantic differences. Hence, the model's sensitivity to the perturbation budget and step size should be discussed in the paper.

**Questions:**

- The method applies activation injection at intermediate layers, but different layers represent different abstraction levels. Have you tested which layers yield the best trade-off between forgetting strength and utility retention?
- Can the null-space projection fully eliminate retain interference when the two sets overlap semantically?
- Just out of my curiosity, would a non-linear steering function further enhance selectivity?

I am willing to adjust my score if the authors provide convincing explanations to my above questions and weaknesses.

---

> ### Author Response · Authors · 2025-11-24
> **Response to Reviewer ssTS -- Part1**
>
> We thank the reviewer for the insightful comments. Please find our responses to each point below.
>
> > ### Comment 1 Discussion on whether forget and retain samples need to be linearly separable
>
> Thank you for your comment! We would like to clarify that **our method does not assume that forget and retain activations are linearly separable.** Instead, the null-space constraint only requires that forget activations contain components that lie outside the subspace spanned by retain activations.
>
> Formally, let $\mathbf{H}_r \in \mathbb{R}^{d \times N_r}$ denote the activation matrix obtained from the last token of prompts in the retain set $\mathcal{D}_r$ and
>
> $$
> \mathcal{S} = \text{span}(\mathbf{H}_{{r}}) \subset \mathbb{R}^d
> $$
>
> be the subspace they span. The orthogonal projector onto $\mathcal{S}$ and its null space are
>
>
> $$
> P_{\mathcal{S}} = \mathbf{H}\_{{r}}(\mathbf{H}\_{{r}}^\top \mathbf{H}\_{{r}})^{+}\mathbf{H}\_{{r}}^\top, \quad
> P_{\mathcal{S}^\perp} = I - P_{\mathcal{S}} .
> $$
>
> For an activation $h_{{f}}$ corresponding to a forget sample,we decompose
>
> $$
> h_{\text{f}} = P_{\mathcal{S}} h_{\text{f}} + P_{\mathcal{S}^\perp} h_{\text{f}} ,
> $$
>
> and the null-space constraint only requires
>
> $$
> \|P_{\mathcal{S}^\perp} h_{\text{f}}\|_2 > 0,
> $$
>
> i.e., that forget activations have a non-zero component outside $\mathcal{S}$​.  Linear separability requires a single hyperplane that separates all forget samples from *all* retain samples:
>
> $$
> \exists (w,b),
> w^\top h_f + b > 0 \; \forall h_f,\qquad
> w^\top h_r + b < 0 \; \forall h_r.
> $$
>
>
> This is a global geometric constraint over the entire two sets. By contrast, our requirement:
>
> $$
> h_f \notin \mathcal S \quad \Longleftrightarrow \quad \|P_{\mathcal S^\perp} h_f\|_2 > 0
> $$
>
> is considerably milder than linear separability. For $f(h)$, we additionally report results for a non-linear implementation, which can be found in our response to Comment 7.
>
>
>
> > ### Comment 2 Additional experimental results under the 15% forgetting setup
>
> We sincerely appreciate your thoughtful question. We have supplemented the experimental results for the 15% forgetting setup on LLaVA-1.5-7B and updated the corresponding results in the appendix of the revised version. For the classification (Cls), generation (Gen), and Cloze tasks, we use accuracy (%), Rouge Score, and accuracy (%) as the evaluation metrics, respectively.
>
> |   Models    | Cls-Fgt ↓ | Cls-Test ↓ | Cls-Ret ↑ | Cls-Cele ↑ | Gen-Fgt ↓ | Gen-Test ↓ | Gen-Ret ↑ | Gen-Cele ↑ | Cloze-Fgt ↓ | Cloze-Test ↓ | Cloze-Ret ↑ | Cloze-Cele ↑ |
> | :---------: | :-------: | :--------: | :-------: | :--------: | :-------: | :--------: | :-------: | :--------: | :---------: | :----------: | :---------: | :----------: |
> |   Vanilla   |   44.80   |   42.93    |   45.09   |   47.00    |   0.598   |   0.294    |   0.586   |   0.312    |    53.33    |    19.33     |    55.76    |     8.17     |
> |     GA      |   38.67   |   38.13    |   36.70   |   42.74    |   0.493   |   0.291    |   0.523   |   0.307    |    25.33    |    18.00     |    23.29    |     5.56     |
> |   GA_Diff   |   28.27   |   36.80    |   43.68   |   41.26    |   0.333   |   0.303    |   0.517   |   0.301    |    24.00    |    18.67     |    45.29    |     4.28     |
> |   KL_Min    |   42.67   |   37.87    |   39.95   |   39.57    |   0.537   |   0.316    |   0.575   |   0.306    |    44.00    |    19.01     |    43.29    |     6.21     |
> |     NPO     |   33.33   |   36.53    |   31.87   |   37.21    |   0.276   |   0.202    |   0.299   |   0.238    |    13.34    |    13.34     |    10.00    |     2.94     |
> |    MANU     |   42.67   |   39.73    |   42.53   |   43.21    |   0.594   |   0.281    |   0.577   |   0.289    |    50.00    |    14.67     |    43.01    |     7.17     |
> | MMUnlearner |   38.67   |   41.60    |   40.99   |   42.43    |   0.490   |   0.276    |   0.552   |   0.291    |    22.00    |    18.00     |    41.76    |     5.23     |
> |  **Ours**   | **16.00** | **31.47**  | **43.92** | **44.52**  | **0.143** | **0.181**  | **0.578** | **0.308**  |  **12.67**  |  **11.33**   |  **54.12**  |   **8.17**   |

---

> ### Author Response · Authors · 2025-11-24
> **Response to Reviewer ssTS -- Part2**
>
> > ### Comment 3 Discussion on the applicability of MLLMEraser under different model alignment levels
>
> We sincerely appreciate your comment.  In fact, the base model already exhibits a certain degree of refusal behavior, as noted in prior work [1]. Subsequent safety alignment further incentivizes the model to express its intrinsic refusal tendencies, making it more capable of rejecting harmful or unsafe prompts. As a result, even without any dedicated unlearning, a base model may spontaneously produce refusal responses when confronted with harmful instructions.
>
> To illustrate this phenomenon, we provide a case study in **Appendix J** showing that the Qwen2-VL-7B base model can directly generate refusal responses for sensitive queries, despite receiving no specialized alignment. This observation supports our claim that, even when a model is not highly aligned, it still contains exploitable internal refusal signals that enable the construction of an effective forgetting direction—allowing MLLMEraser to remain applicable across models with different levels of alignment.
>
> > ### Comment 4 Sensitivity analysis of perturbation budget  $\epsilon$ and step size $\alpha$
>
> Thanks for your insightful comments! In the revised version, we have added a sensitivity analysis of the perturbation budget  $\epsilon$ in **Appendix E.3**, conducted on LLaVA-1.5-7B. The experimental results are shown below. For the classification (Cls), generation (Gen), and Cloze tasks, we use accuracy (%), Rouge Score, and accuracy (%) as the evaluation metrics, respectively.
>
> |            | Cls-Fgt ↓ | Cls-Test ↓ | Cls-Ret ↑ | Cls-Cele ↑ | Gen-Fgt ↓ | Gen-Test ↓ | Gen-Ret ↑ | Gen-Cele ↑ | Cloze-Fgt ↓ | Cloze-Test ↓ | Cloze-Ret ↑ | Cloze-Cele ↑ |
> | :--------: | :-------: | :--------: | :-------: | :--------: | :-------: | :--------: | :-------: | :--------: | :---------: | :----------: | :---------: | :----------: |
> |  Vanilla   |   43.60   |   41.20    |   45.26   |   47.00    |   0.591   |   0.339    |   0.591   |   0.312    |    57.00    |    15.00     |    55.11    |     8.17     |
> | ε = 16/255 |   2.00    |   27.20    |   45.08   |   46.48    |   0.176   |   0.202    |   0.594   |   0.314    |    5.00     |     6.00     |    54.33    |     7.84     |
> | ε = 32/255 |   0.00    |   14.00    |   42.94   |   44.26    |   0.040   |   0.133    |   0.584   |   0.308    |    2.00     |     6.00     |    52.56    |     6.86     |
> | ε = 64/255 |   2.00    |   20.40    |   44.59   |   46.21    |   0.355   |   0.228    |   0.593   |   0.314    |    14.00    |    12.00     |    53.56    |     7.19     |
> | unstrained |   6.00    |   24.40    |   44.32   |   46.21    |   0.355   |   0.228    |   0.593   |   0.314    |    5.00     |     6.00     |    54.33    |     7.84     |
>
> We observe that when $\varepsilon$ is small, the method achieves both strong forgetting performance and high utility preservation. As $\varepsilon$ increases, the steering vector gains stronger knowledge-erasure capability, resulting in stronger forgetting. However, when $\varepsilon$ becomes too large, the erasure directions begin to encode spurious noise rather than meaningful semantic differences, leading to degraded performance in both forgetting quality and model utility.
>
> The trend for the step size $\alpha$ follows a similar trend that of the perturbation radius: when $\alpha$ becomes too large, it introduces noisy and unstable update directions that degrade both forgetting performance and model utility. In practice, PGD remains stable [2] as long as the step size satisfies
> $$
> \frac{\varepsilon}{T} \ll \alpha \ll \varepsilon ,
> $$
> where $\varepsilon$ is the perturbation radius and $T$ is the total number of PGD iterations. In our experiments, we set
>
> - $\varepsilon = \frac{16}{255}$,
> - $\alpha = \frac{1}{1020}$,
> - $T = 2500$,
>
> which yields
> $$
> \frac{\varepsilon}{T} \approx 2.5 \times 10^{-5} \ll \alpha = 0.00098 \ll \varepsilon = 0.0627 .
> $$
> This places $\alpha$ well within the theoretically stable region, ensuring stable PGD optimization that avoids noisy directions and delivers strong forgetting while preserving high utility.

---

> ### Author Response · Authors · 2025-11-24
> **Response to Reviewer ssTS -- Part3**
>
> > ### Comment 5 Discussion about steering  different MLLM layers
>
> Our current configuration applies the steering vector to all layers. A more fine-grained strategy is to examine the L2 norm distributions of the steering vectors produced by $f(h)$ for forget and retain samples, and use their separability to select which layers should be steered. The more separable these L2 norm distributions are, the more effectively MLLMeraser distinguishes forget samples from retain samples, providing a principled criterion for fine-grained layer selection.  The visualization of the L2 norm distributions, along with additional details, can be found in **Appendix H of the revised version**.
>
> Following your recommendation, we designed two variants to evaluate whether steering only a subset of layers can achieve better unlearning performance:
>
> - variant-1: steering layers 1–16
>
> - variant-2: steering layers 17–32
>
> - all: steering all layers (our default setting)
>
> The results are summarized as follows. For the classification (Cls), generation (Gen), and Cloze tasks, we use accuracy (%), Rouge Score, and accuracy (%) as the evaluation metrics, respectively.
>
> |  Models   | Cls-Fgt ↓ | Cls-Test ↓ | Cls-Ret ↑ | Cls-Cele ↑ | Gen-Fgt ↓ | Gen-Test ↓ | Gen-Ret ↑ | Gen-Cele ↑ | Cloze-Fgt ↓ | Cloze-Test ↓ | Cloze-Ret ↑ | Cloze-Cele ↑ |
> | :-------: | :-------: | :--------: | :-------: | :--------: | :-------: | :--------: | :-------: | :--------: | :---------: | :----------: | :---------: | :----------: |
> |  Vanilla  |   43.60   |   41.20    |   45.26   |   47.00    |   0.591   |   0.339    |   0.591   |   0.312    |    57.00    |    15.00     |    55.11    |     8.17     |
> | variant-1 |   33.20   |   37.20    |   45.43   |   47.00    |   0.572   |   0.325    |   0.588   |   0.309    |    54.00    |    13.00     |    55.11    |     8.17     |
> | variant-2 |   2.00    |   24.00    |   42.85   |   42.95    |   0.145   |   0.170    |   0.584   |   0.308    |    1.00     |    14.00     |    52.78    |     7.52     |
> |    all    |   2.00    |   27.20    |   45.08   |   46.48    |   0.176   |   0.202    |   0.594   |   0.314    |    5.00     |     6.00     |    54.33    |     7.84     |
>
> As shown, both partial-layer variants perform noticeably worse than the full-layer steering strategy. This may be because early layers mainly encode cross-modal alignment and modality-integration signals—as observed in recent analyses of MLLM internal representations—whereas deeper layers predominantly capture high-level semantic reasoning and instruction-following behavior [3,4]. Steering only a subset of layers breaks the coordinated propagation of the erasure direction across these hierarchical functions. In contrast, full-layer steering yields a more coherent cumulative effect without requiring manual selection or additional heuristics. Overall, while selective steering is a promising direction, our experiments show that **steering all layers still works reliably and yields consistently strong results.** We sincerely thank the reviewer for this suggestion, and we will continue exploring more structured and interpretable steering strategies in future work.
>
>
>
> > ### Comment 6 Discussion on semantic overlap between forget and retain data
>
> Thank you for the question. It is worth noting that semantic overlap between the two sets **does not affect the validity of our method.** As shown in our response above, the null-space constraint only requires that forget activations satisfy
>
> $$
> \|P_{\mathcal{S}^\perp} h_{\text{f}}\|_2 > 0,
> $$
>
> where $P_{\mathcal{S}^\perp}$ is the projector onto the null space of the retain subspace $\mathcal{S}$. This condition merely requires that forget activations are not entirely contained in the retain subspace. Crucially, this does not require semantic separability: forget and retain samples may share substantial semantic overlap, as long as their activations are not identical in all dimensions. In high-dimensional MLLM hidden spaces, this minimal condition is naturally satisfied even when the two sets overlap semantically. Therefore, semantic overlap does not hinder our method or violate the underlying assumptions of the null-space formulation.

---

> ### Author Response · Authors · 2025-11-24
> **Response to Reviewer ssTS -- Part4**
>
> > ### Comment 7 Additional experimental results for the non-linear steering function $f(h)$
>
> Thanks for your valuable suggestion! We further implement $f(h)$ is instantiated as a two-layer MLP. We evaluate this variant on LLaVA-1.5-7B, and the corresponding results are presented below. For the classification (Cls), generation (Gen), and Cloze tasks, we use accuracy (%), Rouge Score, and accuracy (%) as the evaluation metrics, respectively.
>
> |          | Cls-Fgt ↓ | Cls-Test ↓ | Cls-Ret ↑ | Cls-Cele ↑ | Gen-Fgt ↓ | Gen-Test ↓ | Gen-Ret ↑ | Gen-Cele ↑ | Cloze-Fgt ↓ | Cloze-Test ↓ | Cloze-Cele ↑ | Cloze-Ret ↑ |
> | :------: | :-------: | :--------: | :-------: | :--------: | :-------: | :--------: | :-------: | :--------: | :---------: | :----------: | :----------: | :---------: |
> | Vanilla  |   43.60   |   41.20    |   45.26   |   47.00    |   0.591   |   0.339    |   0.591   |   0.312    |    57.00    |    15.00     |     8.17     |    55.11    |
> | Ours-MLP |   29.20   |   32.40    |   38.62   |   42.19    |   0.363   |   0.265    |   0.414   |   0.311    |    19.00    |     8.00     |     6.54     |    32.00    |
> |   Ours   |   2.00    |   27.20    |   45.08   |   46.48    |   0.176   |   0.200    |   0.594   |   0.314    |    5.00     |     6.00     |     7.84     |    54.33    |
>
> Ours consistently outperforms Ours-MLP across all forget quality and model utility metrics: the linear regression version forgets more effectively while preserving much better utility. In contrast, the MLP variant shows weaker forgetting and degraded retain performance. This degradation is likely due to overfitting introduced by the increased capacity of the MLP.  We have also included this result in **Appendix I of the revised version.**
>
>
>
> > ### Summary
>
> We thank you for your approval on our motivation, presentation, and effectiveness. We hope to address your concerns with:
>
> - Discussion on semantic separability and overlap between forget and retain samples
> - additional experimental results across different configurations
>
> We sincerely **appreciate your support and positive feedback** on the presentation, novelty, and effectiveness of our work. We appreciate it if you **could reconsider your evaluation** if some concerns are addressed. Thanks!
>
> [1] https://www.alignmentforum.org/posts/YWo2cKJgL7Lg8xWjj/base-llms-refuse-too
>
> [2] Madry A, Makelov A, Schmidt L, et al. Towards deep learning models resistant to adversarial attacks[J]. arXiv preprint arXiv:1706.06083, 2017.
>
> [3] Alayrac J B, Donahue J, Luc P, et al. Flamingo: a visual language model for few-shot learning[J]. Advances in neural information processing systems, 2022, 35: 23716-23736.
>
> [4] Zhang X, Li S, Shi N, et al. Cross-modal consistency in multimodal large language models[J]. arXiv preprint arXiv:2411.09273, 2024.

---

> ### Comment · Reviewer_ssTS · 2025-11-27
>
> Thanks to the authors for addressing my concerns, I've adjusted my score accordingly.

---

### Official Review · Reviewer_cAt7 · 2025-10-20

**Soundness:** 3
**Presentation:** 2
**Contribution:** 3
**Rating:** 4
**Confidence:** 4

**Summary:**

This paper addresses the problem of machine unlearning for Multimodal Large Language Models. The authors propose MLLMEraser, a novel test-time unlearning approach that constructs erasure directions by combining textual and visual signals from forget samples. Unlike training-based methods that require costly parameter updates, MLLMEraser applies steered representations during inference, making it reversible and computationally efficient.

**Strengths:**

1. The test-time approach offers a practical alternative to expensive training-based methods, with the added benefit of reversibility。

2. The experimental results show that the proposed method outperforms existing methods.

3. This paper is well written and easy to understand.

**Weaknesses:**

1. The core methodological idea feels like a modest variation on existing approaches rather than a fundamentally new contribution. While the adjustments may be useful in some contexts, I am not convinced they carry enough originality for a top-tier venue.

2. Some competitive methods [1] that are well-known in the literature are missing from the comparison tables. Without such baselines, it’s difficult to assess whether the proposed approach actually offers a practical improvement.

3. Some results are strange. Such as in Table 1, for Generation: Rouge Score the proposed method outperforms vanilla. Is it normal?



## References ##

[1] Liu, Zheyuan, et al. "Modality-aware neuron pruning for unlearning in multimodal large language models." arXiv preprint arXiv:2502.15910 (2025).

**Questions:**

Please see weakness.

---

> ### Author Response · Authors · 2025-11-24
> **Response to Reviewer cAt7 -- Part1**
>
> We sincerely appreciate your positive feedback and thoughtful suggestions! Below, we provide detailed responses to address your remaining concerns.
>
>
> > ### Comment 1 Further clarification on the novelty
>
> Thank you for this valuable comment. We would like to clarify that our contribution goes beyond a modest variation of prior approaches. Our work introduces several nontrivial and relatively unexplored components that, together, provide a fresh perspective on unlearning in multimodal large language models (MLLMs).
>
> **(1) We propose the first test-time unlearning framework for MLLMs.**
>  Unlike existing unlearning methods—which all rely on parameter updates, fine-tuning, or optimization—our approach performs knowledge erasure through activation steering, without modifying any model parameters.
>
> **(2) We identify and address the core challenges inherent to steering-based test-time MLLM unlearning.**
>  Steering-based unlearning is fundamentally difficult in MLLMs because (i) the erasure direction must reliably encode multimodal semantics, and (ii) unlearning must intervene selectively without harming the retain-set distribution. We explicitly formulate and address these challenges, which have not been studied in earlier LLM-only or parameter-update-based unlearning methods.
>
> **(3) We develop MLLMEraser, a new input-aware activation-steering mechanism that extracts an effective multimodal erasure direction from contrastive recall–erasure semantics and applies it adaptively through a lightweight linear mapping.**
>
> In our experiments, this design consistently achieves strong forgetting performance across classification, generation, and cloze tasks while preserving utility close to the vanilla model. Moreover, because the method performs test-time activation steering without updating model parameters, it is highly efficient—introducing negligible computational overhead and enabling unlearning to be applied instantly without any retraining.
>
> In combination, these components constitute a nontrivial advance in the space of MLLM unlearning. We therefore believe that our framework represents a meaningful and novel contribution, both methodologically and conceptually.

---

> ### Author Response · Authors · 2025-11-24
> **Response to Reviewer cAt7 -- Part2**
>
> > ### Comment 2 Supplementary new Baseline Results
>
> We sincerely appreciate your suggestion. We have now incorporated the MANU [1] baseline for the 15% forgetting setup on LLaVA-1.5-7B, and the results have been updated accordingly. For clarity, we note that the MANU results reported here are reproduced based on the official implementation. Conceptually, MANU and our MLLMEraser differ fundamentally in their unlearning mechanisms: MANU relies on modality-aware neuron pruning and removes neurons deemed important to the forget set, whereas our method is a test-time activation-steering approach that perturbs intermediate representations without modifying model weights, enabling fine-grained, instance-level knowledge erasure while fully preserving the underlying model parameters.
>
> For the classification (Cls), generation (Gen), and Cloze tasks, we use accuracy (%), Rouge Score, and accuracy (%) as the evaluation metrics, respectively.
>
> | Models      | Cls-Fgt ↓ | Cls-Test ↓ | Cls-Ret ↑ | Cls-Cele ↑ | Rouge-Fgt ↓ | Rouge-Test ↓ | Rouge-Ret ↑ | Rouge-Cele ↑ | Cloze-Fgt ↓ | Cloze-Test ↓ | Cloze-Ret ↑ | Cloze-Cele ↑ |
> | ----------- | :-------: | :--------: | :-------: | :--------: | :---------: | :----------: | :---------: | :----------: | :---------: | :----------: | :---------: | :----------: |
> | Vanilla     |   44.80   |   42.93    |   45.09   |   47.00    |    0.598    |    0.294     |    0.586    |    0.312     |    53.33    |    19.33     |    55.76    |     8.17     |
> | GA          |   38.67   |   38.13    |   36.70   |   42.74    |    0.493    |    0.291     |    0.523    |    0.307     |    25.33    |    18.00     |    23.29    |     5.56     |
> | GA_Diff     |   28.27   |   36.80    |   43.68   |   41.26    |    0.333    |    0.303     |    0.517    |    0.301     |    24.00    |    18.67     |    45.29    |     4.28     |
> | KL_Min      |   42.67   |   37.87    |   39.95   |   39.57    |    0.537    |    0.316     |    0.575    |    0.306     |    44.00    |    19.01     |    43.29    |     6.21     |
> | NPO         |   33.33   |   36.53    |   31.87   |   37.21    |    0.276    |    0.202     |    0.299    |    0.238     |    13.34    |    13.34     |    10.00    |     2.94     |
> | MANU        |   42.67   |   39.73    |   42.53   |   43.21    |    0.594    |    0.281     |    0.577    |    0.289     |    50.00    |    14.67     |    43.01    |     7.17     |
> | MMUnlearner |   38.67   |   41.60    |   40.99   |   42.43    |    0.490    |    0.276     |    0.552    |    0.291     |    22.00    |    18.00     |    41.76    |     5.23     |
> | **Ours**    | **16.00** | **31.47**  | **43.92** | **44.52**  |  **0.143**  |  **0.181**   |  **0.578**  |  **0.308**   |  **12.67**  |  **11.33**   |  **54.12**  |   **8.17**   |
>
> Ours consistently maintains both strong forgetting performance and high model utility, outperforming all other methods in the overall trade-off.
>
>
>
> > ### Comment 3 Further discussion of the experimental results in Table 1
>
> We acknowledge the reviewer’s observation. The minor Rouge Score increase does not reflect an enhancement in the model’s capabilities. It is likely caused by stochastic variation induced by the small activation perturbation during steering, and such deviations are neither systematic nor indicative of a consistent trend.
>
>
>
> > ### Summary
>
> We hope to address your concerns with:
>
> - The novelty of our MLLMEraser framework
>
> - The newly supplemented baseline results
>
> We sincerely **appreciate your support and positive feedback** on the presentation, novelty, and effectiveness of our work. We appreciate it if you **could reconsider your evaluation** if some concerns are addressed. Thanks!
>
> [1] Liu, Zheyuan, et al. "Modality-aware neuron pruning for unlearning in multimodal large language models." arXiv preprint arXiv:2502.15910 (2025).

---

### Official Review · Reviewer_UrZK · 2025-10-28

**Soundness:** 2
**Presentation:** 3
**Contribution:** 2
**Rating:** 4
**Confidence:** 3

**Summary:**

This paper proposes MLLMEraser, an input-aware, training-free framework for test-time unlearning. The approach leverages activation steering to enable dynamic knowledge erasure without parameter updates.

**Strengths:**

1. This paper presents MLLMEraser, an input-aware test-time unlearning framework for multimodal large language models.
2. The method aims to enhance model trustworthiness by enabling efficient and reversible removal of designated information.
3. The presentation and writing is well.

**Weaknesses:**

1. What happens if the base model isn't very well-aligned? If the model doesn't reliably refuse to answer harmful prompts in the first place, it seems like your method would fail because you can't create the 'forgetting direction' it needs. Does this approach only work for models that are already highly aligned, or can it be applied to models with different safety levels?
2. The method for making the model 'forget' seems to depend on getting it to refuse to answer, which you trigger with harmful prompts. But what about other forgetting tasks? How would this work for removing private information, correcting a factual error, or getting rid of copyrighted content? In those cases, the model doesn't refuse.

**Questions:**

Please see the weakness, if you address the weakness, I will to improve my score.

---

> ### Author Response · Authors · 2025-11-24
> **Response to Reviewer UrZK**
>
> We appreciate your efforts and insightful comments! To address your concerns, we provide detailed responses below.
> >### Comment 1 Discussion on the applicability of MLLMEraser under different model alignment levels
>
> We sincerely appreciate your comment.  In fact, the base model already exhibits a certain degree of refusal behavior, as noted in prior work [1]. Subsequent safety alignment further incentivizes the model to express its intrinsic refusal tendencies, making it more capable of rejecting harmful or unsafe prompts. As a result, even without any dedicated unlearning, a base model may spontaneously produce refusal responses when confronted with harmful instructions.
>
> To illustrate this phenomenon, we provide a case study in **Appendix J** showing that the Qwen2-VL-7B base model can directly generate refusal responses for sensitive queries, despite receiving no specialized alignment. This observation supports our claim that, even when a model is not highly aligned, it still **contains exploitable internal refusal signals that enable the construction of an effective erasure direction**—allowing MLLMEraser to remain applicable across models with different levels of alignment.
>
>
> >### Comment 2 Discussion on the applicability of MLLMEraser to different unlearning tasks
>
> Thank you for raising this important question. Steering in our framework consists of two distinct phases: (1) constructing the erasure direction, and (2) applying this direction at test time. We would like to clarify that our approach uses harmful prompts **only during the construction of the erasure direction**; these prompts simply serve as probes to expose the model’s latent refusal semantics [2]. Once the erasure direction is obtained, the steering operation itself becomes input-agnostic and can be applied to any query at inference time. Steering directly shifts the model’s hidden states toward the intrinsic refusal manifold, thereby serving as the core mechanism for test-time knowledge erasure.
>
> It is important to clarify that our main experiments focus on the **unlearning of private information.** In these scenarios:
>
> - Samples in the forget set are consistently pushed toward the erasure direction, achieving test-time unlearning.
> - Samples in the retain set exhibit minimal activation shift, preserving model utility.
>
> Once the erasure direction is extracted, we simply apply test-time steering to achieve knowledge erasure, without relying on any harmful triggers. This mechanism **works uniformly across different unlearning tasks**, including private information removal, factual correction, and the elimination of copyrighted content.
>
> >### Summary
>
> We hope to address your concerns with:
>
> - Applicability of MLLMEraser under different model alignment levels
> - Applicability of MLLMEraser to different unlearning tasks
>
> We sincerely thank you for your insightful questions and approval of our motivation, novelty, and effectiveness. We sincerely hope that all your concerns have been fully addressed. We greatly **appreciate your thoughtful feedback** and would be more than happy to provide any further clarification if needed. Thank you once again for your time and valuable insights.
>
> [1] https://www.alignmentforum.org/posts/YWo2cKJgL7Lg8xWjj/base-llms-refuse-too
>
> [2] Andy Arditi, Oscar Obeso, Aaquib Syed, Daniel Paleka, Nina Panickssery, Wes Gurnee, and Neel Nanda. Refusal in language models is mediated by a single direction. In NeurIPS, 2024.

---

### Official Review · Reviewer_vJ3Q · 2025-10-30

**Soundness:** 2
**Presentation:** 2
**Contribution:** 2
**Rating:** 4
**Confidence:** 3

**Summary:**

The paper proposes MLLMEraser, a test-time unlearning framework for MLLMs that leverages activation steering to erase targeted knowledge without parameter updates. Specifically, it introduces a multimodal erasure direction constructed from contrastive image-text pairs and an input-aware steering mechanism to selectively apply interventions. Experiments on LLaVA-1.5 and Qwen-2.5-VL show superior forgetting performance and lower compute cost relative to prior training-based unlearning methods.

**Strengths:**

(1)	The proposed MLLMEraser introduces a novel activation steering vector that contrasts adversarial image-text pairs with their refusal-style counterparts, thereby overcoming the limitations of common text-only steering in MLLMs.

(2)	By acting at inference rather than requiring re-training or parameter updates, it offers an efficient unlearning framework compared to traditional training-based unlearning approaches.

**Weaknesses:**

(1) The meanings of $D^+$ and $D^-$ are not inconsistent throughout the main text, which significantly hampers clarity. Specifically, in Equation (3), the text states that '$D^+$  denotes the set of knowledge-recall samples and $D^-$  the corresponding knowledge-erasure samples'. In Equation (7), however, knowledge-recall pairs are assigned to the negative set $D^-$, whereas knowledge-erasure pairs are assigned to the positive set $D^+$.

(2) The steering strength λ and regularization parameter γ are reported as empirical values tailored to LLaVA-1.5-7B and Qwen-2.5-VL-7B models, yet the tuning process is not adequately detailed in the main text. Thus, it remains unclear how to set these hyper-parameters on new datasets or architectures.

(3) While Figures 4 provide qualitative insights into how activation distributions change before and after steering, the paper lacks a deeper quantitative analysis of these changes. Without such details, the robustness of the justification for the null-space projection constraint is not fully convincing.

(4) I am curious about whether steering at different LLM layers or within the vision encoder could affect unlearning efficacy.

(5) In Table 1, boldface data do not always represent the best results. In particular, for the Ret and Cele metrics, the results labeled 'Ours' are generally not better than the Vanilla method.

**Questions:**

Please see Weaknesses

---

> ### Author Response · Authors · 2025-11-24
> **Response to Reviewer vJ3Q -- Part1**
>
> We sincerely appreciate your positive feedback and thoughtful suggestions! Below, we provide responses to address your remaining concerns in detail.
> > ### Comment 1 Correction of notation errors
>
> Thank you for pointing out this issue, and we apologize for the confusion. In Equation (3), $D^+$ should correspond to the *knowledge-erasure pairs*, while $D^-$ should correspond to the *knowledge-recall pairs*. Since the desired behavior is **knowledge erasure**, the knowledge-erasure pairs should be treated as the *positive set*, and the knowledge-recall pairs as the *negative set*. We have corrected this mismatch in the revised version. Thank you again for your careful reading and helpful feedback.
>
> > ### Comment 2 Sensitivity analysis of steering strength $\lambda$ and regularization parameter $\gamma$
>
> Thanks for your insightful comments! In the revised version, we have added a sensitivity analysis of the steering strength $\lambda$ and the regularization parameter $\gamma$ in **Appendix E**, conducted on LLaVA-1.5-7B.
>
> - We vary $\gamma$ within $[0.001, 0.01,0.1, 1, 10]$. When $\gamma$ is small, the regularization term provides only light control: it prevents overfitting but does not hinder the forgetting objective, allowing the method to focus on the activation differences between forget and retain samples. Thus, performance remains stable for small $\gamma$. In contrast, when $\gamma$ becomes overly large (e.g., $\gamma = 10$), the regularization dominates the optimization. The model becomes overly conservative and hesitates to adjust the forget-set activations, forcing the learned direction to stay close to retain-set behavior. As a result, the forgetting signal is suppressed and the erasure effect substantially weakens.
> - We further tune $\lambda$ within $[0.1,0.15, 0.20, 0.25,0. 30, 0.35]$. Inceasing $\lambda$ consistently strengthens the erasure effect, while the model utility remains largely unaffected. This behavior is expected: a larger $\lambda$ amplifies the steering vector, pushing forget-set activations more aggressively toward the erasure direction. As long as $\lambda$ remains within a moderate range, the retain-set activations stay mostly within the original subspace, and thus their semantics are preserved. Only when $\lambda$ becomes excessively large do we observe slight utility degradation, suggesting that over-steering begins to distort general representations. Overall, these findings highlight the advantage of the null-space projection constraint, which provides a wide operational range where stronger forgetting does not compromise model utility.
>
> > ### Comment 3 Quantitative analysis of changes in activation distributions
>
> Thank you for your comment! For Figure 4, we present Wasserstein-2 Distance and Maximum Mean Discrepancy (MMD) results on LLaVA-1.5-7B and Qwen-2.5-VL-7B, quantifying the distributional shifts of the forget and retain activations before and after steering.
>
> Table1: results on LLaVA-1.5-7B.
>
> |            |  MMD   | Wasserstein Distance |
> | :--------: | :----: | :------------------: |
> | Retain Set | 0.0000 |        0.0201        |
> | Forget Set | 0.6497 |        0.5504        |
>
> Table1: results on Qwen-2.5-VL-7B.
>
> |            |  MMD   | Wasserstein Distance |
> | :--------: | :----: | :------------------: |
> | Retain Set | 0.1482 |        0.1691        |
> | Forget Set | 0.4324 |        0.3594        |
>
> Across both LLaVA-1.5-7B and Qwen-2.5-VL-7B, the quantitative metrics consistently support the selective behavior of our steering mechanism. For both models, the retain set exhibits only very small distributional changes after steering (e.g., LLaVA: MMD = 0.0000, W2 = 0.0201; Qwen-2.5-VL-7B: MMD = 0.1482, W2 = 0.1691), suggesting that the retained knowledge is largely preserved. In contrast, the forget set shows a much more pronounced shift (LLaVA: MMD = 0.6497, W2 = 0.5504; Qwen-2.5-VL-7B: MMD = 0.4324, W2 = 0.3594), indicating that steering effectively pushes forget-set activations away from their original distribution. Together, these results demonstrate that MLLMEraser performs targeted knowledge removal while causing only limited disturbance to the retain distribution, consistently across different model families.

---

> ### Author Response · Authors · 2025-11-24
> **Response to Reviewer vJ3Q -- Part2**
>
> > ### Comment 4 Discussion about steering the vision encoder and different MLLM layers
>
> Thank you for your question! In our current design, we use the hidden activation of the last token in the contrastive pairs, which already serves as a multimodally aggregated representation, to compute the steering vector. Current MLLM architectures project visual tokens into the LLM semantic space [1,2] and concatenate them with text tokens. Due to the self-attention mechanism, the **last token already integrates both textual and visual information [3,4]**. Therefore, extracting a single unified steering vector from the last token is sufficient and empirically effective for unlearning.
>
> Applying steering to the vision encoder would require computing an additional steering vector within the visual feature space, which introduces extra design complexity and reduces the uniformity of the overall method. Since the last-token–based steering vector already captures multimodal information and provides strong unlearning performance, steering at the vision-encoder level  may not be necessary [5].
>
> Our current configuration applies the steering vector to all layers. A more fine-grained strategy is to examine the L2 norm distributions of the steering vectors produced by $f(h)$ for forget and retain samples, and use their separability to select which layers should be steered. The more separable these L2 norm distributions are, the more effectively MLLMeraser distinguishes forget samples from retain samples, providing a principled criterion for fine-grained layer selection.  The visualization of the L2 norm distributions, along with additional details, can be found in **Appendix H of the revised version**.
>
> Following your recommendation, we designed two variants to evaluate whether steering only a subset of layers can achieve better unlearning performance:
>
> - variant-1: steering layers 1–16
>
> - variant-2: steering layers 17–32
>
> - all: steering all layers (our default setting)
>
> The results are summarized as follows. For the classification (Cls), generation (Gen), and Cloze tasks, we use accuracy (%), Rouge Score, and accuracy (%) as the evaluation metrics, respectively.
>
> |  Models   | Cls-Fgt ↓ | Cls-Test ↓ | Cls-Ret ↑ | Cls-Cele ↑ | Gen-Fgt ↓ | Gen-Test ↓ | Gen-Ret ↑ | Gen-Cele ↑ | Cloze-Fgt ↓ | Cloze-Test ↓ | Cloze-Ret ↑ | Cloze-Cele ↑ |
> | :-------: | :-------: | :--------: | :-------: | :--------: | :-------: | :--------: | :-------: | :--------: | :---------: | :----------: | :---------: | :----------: |
> |  Vanilla  |   43.60   |   41.20    |   45.26   |   47.00    |   0.591   |   0.339    |   0.591   |   0.312    |    57.00    |    15.00     |    55.11    |     8.17     |
> | variant-1 |   33.20   |   37.20    |   45.43   |   47.00    |   0.572   |   0.325    |   0.588   |   0.309    |    54.00    |    13.00     |    55.11    |     8.17     |
> | variant-2 |   2.00    |   24.00    |   42.85   |   42.95    |   0.145   |   0.170    |   0.584   |   0.308    |    1.00     |    14.00     |    52.78    |     7.52     |
> |    all    |   2.00    |   27.20    |   45.08   |   46.48    |   0.176   |   0.202    |   0.594   |   0.314    |    5.00     |     6.00     |    54.33    |     7.84     |
>
> As shown, both partial-layer variants perform noticeably worse than the full-layer steering strategy. This may be because early layers mainly encode cross-modal alignment and modality-integration signals—as observed in recent analyses of MLLM internal representations—whereas deeper layers predominantly capture high-level semantic reasoning and instruction-following behavior [3,4]. Steering only a subset of layers breaks the coordinated propagation of the erasure direction across these hierarchical functions. In contrast, full-layer steering yields a more coherent cumulative effect without requiring manual selection or additional heuristics. Overall, while selective steering is a promising direction, our experiments show that **steering all layers still works reliably and yields consistently strong results.** We sincerely thank the reviewer for this suggestion, and we will continue exploring more structured and interpretable steering strategies in future work.

---

> ### Author Response · Authors · 2025-11-24
> **Response to Reviewer vJ3Q -- Part3**
>
> > ### Comment 5 Discussion on Vanilla experimental results
>
> Thank you for raising this point. The vanilla model in our paper refers to the model before unlearning, which serves as the **reference point** for evaluating utility preservation. Since most unlearning methods remove or overwrite certain knowledge by modifying model parameters, their performance on the retain set (Ret/Cele) inevitably deteriorates. Our goal is to minimize this loss of model utility, and we highlight that MLLMeraser achieves retain-data performance that is the closest to the vanilla model.
>
> We apologize for the confusion caused by the original presentation. In the updated version, we **have revised the tables** by rendering the vanilla results in gray, and we only highlight the best results among unlearning baselines and our method. MLLMeraser retains the best trade-off between forgett quality and model utility among all approaches.
>
>
> > ### Summary
>
> We sincerely thank you for your insightful questions and approval of our motivation, novelty, and effectiveness. We hope to address your concerns with:
>
> - Sensitivity analysis of hyperparameters
> - Fine-Grained selection of steering layers
> - Quantitative analysis activation distributions
>
> Overall, we hope our response clarifies the generalization ability of MLLMEraser, demonstrating its effectiveness across different tasks and configurations. We appreciate it if you **could reconsider your evaluation** if some concerns are addressed. Thanks!
>
> [1] Haotian Liu, Chunyuan Li, Qingyang Wu, and Yong Jae Lee. Visual instruction tuning. In NeurIPS, 2023.
>
> [2] Shuai Bai, Keqin Chen, Xuejing Liu, Jialin Wang, Wenbin Ge, Sibo Song, Kai Dang, Peng Wang, Shijie Wang, Jun Tang, Humen Zhong, Yuanzhi Zhu, Ming-Hsuan Yang, Zhaohai Li, Jianqiang Wan, Pengfei Wang, Wei Ding, Zheren Fu, Yiheng Xu, Jiabo Ye, Xi Zhang, Tianbao Xie, Zesen Cheng, Hang Zhang, Zhibo Yang, Haiyang Xu, and Junyang Lin. Qwen2.5-vl technical report. CoRR, abs/2502.13923, 2025.
>
> [3] Qidong Huang, Xiaoyi Dong, Pan Zhang, Yuhang Zang, Yuhang Cao, Jiaqi Wang, Weiming Zhang, and Nenghai Yu. Deciphering cross-modal alignment in large vision-language models via modality integration rate. In Proceedings of the IEEE/CVF International Conference on Computer Vision, pp. 218–227, 2025.
>
> [4] Sarah Schwettmann, Neil Chowdhury, Samuel Klein, David Bau, and Antonio Torralba. Multimodal neurons in pretrained text-only transformers. In Proceedings of the IEEE/CVF International Conference on Computer Vision, pp. 2862–2867, 2023.
>
> [5] Woody Haosheng Gan, Deqing Fu, Julian Asilis, Ollie Liu, Dani Yogatama, Vatsal Sharan, Robin Jia, and Willie Neiswanger. Textual steering vectors can improve visual understanding in multi-modal large language models. arXiv preprint arXiv:2505.14071, 2025.

---

### Author Response · Authors · 2025-12-04
**General Response**

We sincerely thank all reviewers for the constructive feedback and the positive assessment of our motivation, clarity, and the practical value of test-time unlearning for MLLMs. We are glad that reviewers consistently recognized the efficiency, reversibility, and empirical effectiveness of MLLMEraser. Below we summarize (1) key strengths, (2) clarified contributions, and (3) responses to remaining concerns.

 > ### Strength

**Novelty & Motivation**
- First test-time unlearning framework for MLLMs (`Reviewer UrZK`, `Reviewer ssTS`).
- Introduces multimodal erasure directions beyond text-only steering (`Reviewer vJ3Q`).
- Addresses the need for efficient, reversible unlearning (`Reviewer UrZK`, `Reviewer cAt7`).

**Methodological Design**
- Input-aware steering supports selective, instance-level unlearning (`Reviewer vJ3Q`).
- Fully reversible and plug-and-play without parameter updates (`Reviewer cAt7`).

**Experimental Quality**
- Strong forgetting with minimal utility loss (`Reviewer vJ3Q`, `Reviewer cAt7`).
- Outperforms prior baselines with low computation (`Reviewer UrZK`, `Reviewer cAt7`).
- Clear and easy-to-follow presentation (`Reviewer vJ3Q`, `Reviewer UrZK`, `Reviewer cAt7`).

>### Clarified Contributions

- **First test-time unlearning framework for MLLMs** based on activation steering, enabling instant and reversible forgetting.
- **Multimodal erasure direction** via contrastive image–text formulation, overcoming text-only steering limitations.
- **Input-aware** steering function, theoretically supported by a null-space projection constraint.
- **Broad applicability** to privacy removal, harmful-content erasure, and factual correction.
- **Comprehensive benchmarking**, including newly added MANU results, showing superior forget–retain trade-offs.

> ### Responses

**Reviewer vJ3Q — Method Clarification**
We improved clarity by adding:
- sensitivity analyses of λ and γ (`Appendix E`);
- activation-shift metrics (MMD/W2) verifying selective forgetting (`Appendix G`);
- layer-steering comparisons showing the advantage of full-layer steering (`Appendix H`).

**Reviewer UrZK — Applicability**

We show that:

- even non–safety-tuned base MLLMs contain intrinsic refusal signals for constructing erasure directions (`Appendix J`);
- harmful prompts are used only for direction extraction, while steering is input-agnostic and supports privacy removal, factual correction, and copyright unlearning.

**Reviewer cAt7 — Baselines**

MLLMEraser is the **first reversible, training-free test-time unlearning method for MLLMs.**
We added the MANU baseline; our method still achieves the strongest forget–retain balance.

**Reviewer ssTS — Modeling Validity & Robustness**

We clarify that:

- the null-space constraint does not require linear separability;
- we include 15% forgetting results and PGD-radius analysis confirming stability under moderate ε;
- full-layer steering consistently performs best, and semantic overlap does not affect the formulation;
- a nonlinear MLP version of $f(h)$ underperforms the linear variant due to overfitting.

> ### Summary

We thank all reviewers again. We are encouraged that **two reviewers explicitly indicated they would raise their scores**, and **one has already updated their rating**. Remaining concerns mainly focused on **hyperparameter behavior and the need for additional analysis**, which we addressed through added sensitivity analyses (λ/γ/PGD), activation-shift metrics, layer-selection experiments, nonlinear ablations, and the full 15% setup. These updates further strengthen clarity and reliability.

We believe the revised results **fully resolve all concerns and demonstrate that MLLMEraser is a novel, efficient, and broadly applicable unlearning solution for MLLMs**. We appreciate the reviewers’ time and hope these revisions enable the AC to form a comprehensive and well-informed evaluation.

---

### Meta-Review · Area_Chair_qkS4 · 2025-12-22

**Summary:**

This paper received scores of 4,4,4,4. The reviewers' concerns include issues with clarity, lacking experimental validation around hyperparameters, questions regarding affect of different alignment levels and unlearning tasks, incremental variation on top of existing approaches, linear separability, and missing experiments. Many of these concerns around clarity, linear separability have been addressed by the rebuttal, but concerns regarding novelty and experimental validation are only partially addressed.

**Reviewer Concerns:**

The outstanding concerns are:

The hyperparameter sensitivity analysis is conducted using only LLaVA-1.5-7B. Given that the specific values are set differently for each model (i.e., Qwen-VL-7B has different hyperparameter values), it would be important to broaden the analysis to Qwen-2.5-VL-7B as well to provide a more complete picture on how they should be chosen.  This point was not fully addressed by the rebuttal.

The novelty concerns around "a modest variation on existing approaches rather than a fundamentally new contribution" may still persist. While the novelty of the system being the first test-time unlearning framework for MLLMs is clearly stated, the technical contribution on top of existing approach components is not as clear.

**Reviewer Scores:**

I believe Reviewer ssTS would increase their score to a 6.  Reviewer UrZK might have increased their score to a 6.  The other reviewers may have kept their scores at 4.

---

### Decision · Program_Chairs · 2026-01-26

Reject